# OnlineSplatter: Pose-Free Online 3D Reconstruction for Free-Moving Objects

**Mark He Huang**[1,3], **Lin Geng Foo**[2], **Christian Theobalt**[2], **Ying Sun**[3], **De Wen Soh**[1]

[1]Singapore University of Technology and Design, Singapore
[2]Max Planck Institute for Informatics, Saarland Informatics Campus, Germany
[3]Institute for Infocomm Research (I[2]R) & Centre for Frontier AI Research, A*STAR, Singapore
`he_huang@mymail.sutd.edu.sg`

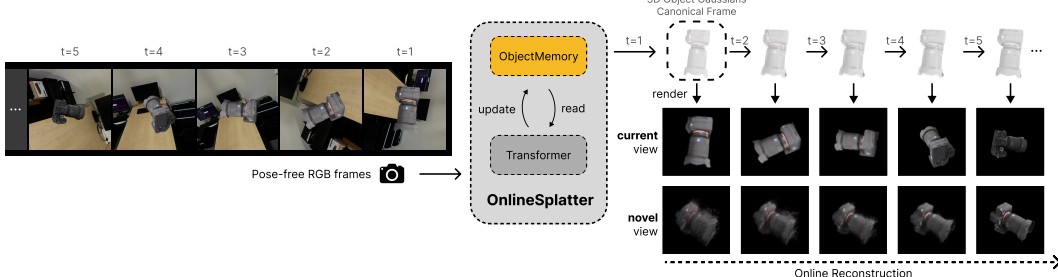

Figure 1: Outline of proposed **OnlineSplatter**. From the incoming stream of pose-free RGB frames, OnlineSplatter "splats" the observations into a canonical cloud of 3D Gaussians anchored to the first frame. Every new frame triggers a single, $O(1)$ **memory-and-time** update that immediately improves the reconstruction of the object. The system copes seamlessly with **freely moving objects** and requires neither pre-computed poses nor depth maps. The result is a continually improving 3D representation suitable for real-time applications.

## Abstract

Free-moving object reconstruction from monocular video remains challenging, particularly without reliable pose or depth cues and under arbitrary object motion. We introduce OnlineSplatter, a novel online feed-forward framework generating high-quality, object-centric 3D Gaussians directly from RGB frames without requiring camera pose, depth priors, or bundle optimization. Our approach anchors reconstruction using the first frame and progressively refines the object representation through a dense Gaussian primitive field, maintaining constant computational cost regardless of video sequence length. Our core contribution is a dual-key memory module combining latent appearance-geometry keys with explicit directional keys, robustly fusing current frame features with temporally aggregated object states. This design enables effective handling of free-moving objects via spatial-guided memory readout and an efficient sparsification mechanism, ensuring comprehensive yet compact object coverage. Evaluations on real-world datasets demonstrate that OnlineSplatter significantly outperforms state-of-the-art pose-free reconstruction baselines, consistently improving with more observations while maintaining constant memory and runtime. Project page: `markhh.com/OnlineSplatter`

# 1 Introduction

Real-time 3D reconstruction of freely moving objects from monocular video remains a fundamental challenge in computer vision, with far-reaching implications for robotics, augmented reality, and interactive 3D applications [31, 33, 34, 45, 41, 53]. Although recent work achieves impressive results on static scenes, real-world deployments are often more demanding: objects move freely, undergoing arbitrary rotations and translations while a moving camera observes them. This dynamic setting violates the static-scene assumptions, reliable pose or depth cues, that underpin most existing methods. The challenge is especially acute in an online setting, where systems must update their understanding of objects as every new frame arrives, a capability essential for autonomous robots and AR devices operating in unpredictable environments.

Recent advances in 3D object reconstruction follow two main paradigms. The first leverages diffusion-based generative models and large-scale object-level priors [11, 23, 36, 22, 21, 25, 38] to synthesize plausible 3D assets from single or a few images. While generating visually compelling geometry, these methods rely on their learned priors to hallucinate unseen parts of the object. Although beneficial for 3D asset generation, this approach limits their applicability in real-time perception. The second paradigm comprises pointmap-based feed-forward methods [42, 8, 24, 48, 39, 2, 51] that directly regress pixel-aligned pointmaps from unposed images. Although effective for stationary scenes and accurate surface recovery, they falter when confronted with objects undergoing unrestricted motion.

A common thread among existing approaches is their reliance on camera poses, depth information, or global optimization [20, 9, 31, 44]. Such limitations not only restrict their applicability in online settings but also hinder their performance in environments where additional sensor data is unavailable.

Motivated by these challenges, we propose *OnlineSplatter*, a feed-forward framework for online reconstruction of freely moving objects. Anchoring reconstruction in the canonical coordinate system defined by the first frame, our network predicts a dense field of per-pixel Gaussian primitives and refines the object model causally as new RGB frames arrive (Fig. 1). Moreover, to control memory growth as observations accumulate, we propose an attention-based memory module that fuses incoming frame features with a compact latent state, eliminating the overhead of bundle adjustment or additional data processing. Central to this design is a dual-key memory retrieval strategy that combines appearance-geometry features with explicit spatial cues, providing robust temporal fusion and comprehensive spatial coverage.

Our contributions are: (i) a novel feed-forward framework for object-centric online 3D reconstruction that operates on monocular RGB streams in real-time, eliminating the need for camera poses, depth priors, or global optimization while maintaining constant computational complexity regardless of sequence length; (ii) a dual-key memory module that pairs latent features with spatial cues for efficient spatial-temporal fusion and principled sparsification; (iii) extensive experiments on GSO [7] and HO3D [10] demonstrate state-of-the-art performance on freely moving objects, paving the way for practical online reconstruction in dynamic environments.

# 2 Related Works

**Pose-free 3D Reconstruction.** Eliminating camera pose as input remains a key challenge in 3D reconstruction. Classical pipelines such as COLMAP [30] recover structure and poses via incremental SfM with MVS. Recent methods like 3D Gaussian Splatting [16], when combined with joint-pose optimization [14, 12, 17], can handle unposed images but still require static scenes. These optimization-based approaches fail to reconstruct moving objects due to their reliance on cross-frame correspondences, even when object masks are available. In contrast, our approach processes each frame causally in a feed-forward manner, eliminating the need for cross-frame optimization while handling freely moving objects. Recent feed-forward methods like DUSt3R [42] predict aligned pointmaps from pairs of unposed images within a shared coordinate space and merge them via global alignment. While this avoids explicit optimization, DUSt3R and its variants (e.g., Spann3R [39], VGGT [40], etc. [8, 51]) still assume static scenes and treat moving objects as outliers due to low cross-frame consistency. Their implicit reliance on large background surfaces further limits their applicability in dynamic, online reconstruction environments. Our method differs by focusing solely on object reconstruction without requiring background information, making it suitable for online reconstruction of freely moving objects.

**Optimization-based Object Reconstruction.** Optimization-based methods typically require global bundle adjustment across all frames, which poses fundamental limitations for real-time applications. Early approaches such as BARF [20] employ a coarse-to-fine registration strategy to jointly optimize shape and pose for NeRF using unposed RGB frames. Similarly, Hampali *et al.* [9] propose a splitting-and-merging strategy tailored for in-hand object scanning, enabling reconstruction of freely moving objects. However, both methods rely on global bundle optimization over all frames, rendering them unsuitable for online, causal settings. Fmov [31] leverages a virtual camera model to narrow the camera pose search space during initial joint 3D shape and pose optimization, followed by global optimization across frames. BundleSDF [44] utilizes a keypoint matching network and explicit frame selection strategy to optimize an object-centric neural signed distance field in a causal manner. While it can handle freely moving objects, it requires ground truth depth information as input during optimization. In contrast, our method employs an attention-based memory module that combines relative pose and latent features for purely feed-forward reconstruction, eliminating the need for depth input and computationally expensive cross-frame optimization.

**Few-view Object Reconstruction.** Recent advances in few-view reconstruction focus on generating 3D assets from limited views. Earlier methods like FvOR [50] combine learned object pose priors with alternating pose-shape optimization to reconstruct unknown objects from just a few images. FORGE [13] proposes a framework that jointly infers relative camera poses and fuses per-view 3D voxel features into a neural radiance field, achieving category-agnostic object reconstruction. However, these methods implicitly assume fully visible, centered objects and struggle with occlusions, sub-optimal observations, or dense view sequences. More recent large-scale data-driven approaches [36, 11, 54, 35, 46, 38, 37, 25, 56, 47, 23, 22] generate plausible geometry but typically focus on single-shot reconstruction and do not incorporate temporal observations for incremental refinement. In contrast, our framework continuously updates its representation as new frames arrive, yielding high-fidelity reconstructions that improve over time while remaining real-time and pose-free.

# 3 Method

The goal of our method is to perform an online reconstruction of a freely moving rigid object using monocular RGB images without relying on known camera poses or any object prior (such as depth, shape, or category). The term "online" implies that our approach processes incoming data causally, updating the reconstructed object representation incrementally as new frames become available. Fig. 2 shows an overview of our proposed OnlineSplatter framework. In the following sections, we elaborate on the details of our framework's pipeline (Sec. 3.1), the design of our dual-key 3D Object Memory (Sec. 3.2), and the training and inference procedure for our model (Sec. 3.3).

## 3.1 OnlineSplatter Pipeline

**Image Encoding.** At each timestep $t$, we observe an RGB view $V_t$. While most reconstruction methods [42] implicitly rely on static scenes and background surfaces for stability, our approach explicitly handles freely moving objects in isolation. To focus on the object of interest, we leverage an off-the-shelf video object segmentation model XMem [3] to obtain the object mask $M_t$, and apply it to each frame to remove the background. The masked frame $V_t'$ is then encoded into patch features via a hybrid strategy that concatenates two complementary encoders:

$$f_{vt} = \text{Concat}(\text{Encoder}_1^I(V_t'), \text{Encoder}_2^I(V_t')) \tag{1}$$

where $\text{Encoder}_1^I$ is a frozen DINO backbone [1, 28, 32], providing strong self-supervised appearance cues. However, DINO alone lacks the 3D awareness required for accurate reconstruction. We therefore introduce a learnable counterpart, $\text{Encoder}_2^I$, which adopts the same architecture but is trained end-to-end within OnlineSplatter to capture complementary geometric cues. This dual-encoder design enables our model to leverage both rich visual priors and geometry-aware features.

**OnlineSplatter Transformer.** After encoding the input image at each timestep, these features are input into our proposed OnlineSplatter Transformer, which is designed to process three distinct types of tokenized inputs: *reference-view tokens* $\mathbf{T}_{ref}^{in}$ (encoded from the initial frame $V_0$), *source-view tokens* $\mathbf{T}_{src,t}^{in}$ (encoded from the current frame $V_t$), and *memory-readout tokens* $\mathbf{T}_{mem,t}^{in}$ (readout from the object memory at time $t$). To differentiate and contextualize these tokens, we introduce

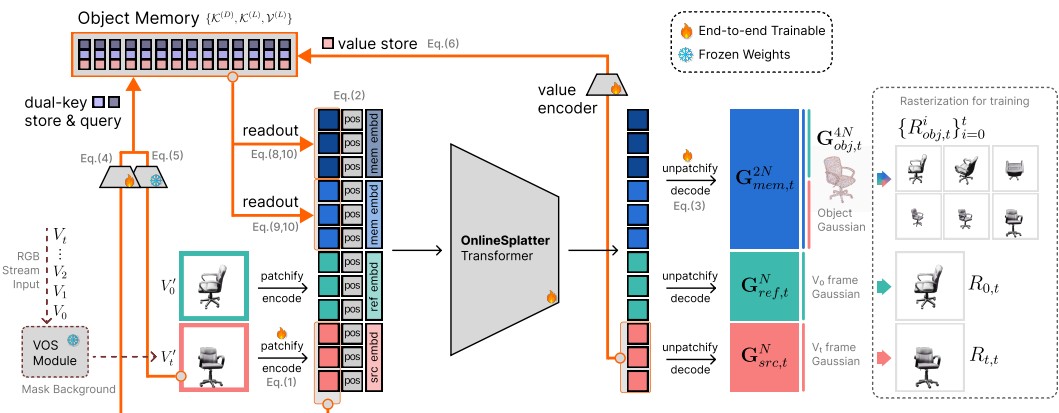

Figure 2: Overview of *OnlineSplatter* Pipeline. The input to our framework consists of a stream of RGB images $\{V_t\}_{t=0}^{N}$, where object masks $\{M_t\}_{t=0}^{N}$ are generated and applied to remove background on-the-fly using an off-the-shelf online video segmentation (OVS) module running alongside our framework. At each timestep $t$, *OnlineSplatter* processes the input frame $V_t$ by first patchifying it into patch tokens. These tokens are then fed into a transformer-based architecture, which directly reasons and outputs pixel-aligned 3D Gaussian representations in a canonical space. Central to our method is object memory, an implicit module based on cross-attention, which is queried and updated at every timestep. This memory enables the incremental reconstruction of the object, consistently refining the object representation ($\mathbf{G}_{obj,t}^{4N}$) as new observations arrive in a fully feed-forward manner.

corresponding learnable embeddings to each of the tokenized features. These embeddings are added with their respective encoded features and positional embeddings prior to transformer processing.

$$\mathbf{T}_{view}^{in} = f_{view} + f_{pos}^{emb} + f_{view}^{emb}, \quad \forall view \in \{ref, src, mem\} \tag{2}$$

**Gaussian Decoding.** The three types of token $\{\mathbf{T}_{ref}^{in}, \mathbf{T}_{src,t}^{in}, \mathbf{T}_{mem,t}^{in}\}$ are processed jointly by our unified OnlineSplatter Transformer that produces the output tokens $\{\mathbf{T}_{ref}^{out}, \mathbf{T}_{src,t}^{out}, \mathbf{T}_{mem,t}^{out}\}$, which are then decoded into 3D Gaussian parameters through a trainable unpatchifier.

$$\mathbf{G}_{obj,t}^{4N} := \{\mathbf{G}_{mem,t}^{2N}, \mathbf{G}_{ref,t}^{N}, \mathbf{G}_{src,t}^{N}\} = \text{Unpatchify}(\{\mathbf{T}_{mem,t}^{out}, \mathbf{T}_{v_0}^{out}, \mathbf{T}_{vt}^{out},\}) \tag{3}$$

where each Gaussian set $\mathbf{G}^K$ consists of $K$ Gaussian primitives defined as: $\{\boldsymbol{\mu}_k, \mathbf{r}_k, \mathbf{s}_k, \mathbf{c}_k, o_k\}_{k=1}^{K}$ with $N = H \times W$, corresponding to $N$ pixels per RGB input $V \in \mathbb{R}^{H \times W \times 3}$. Note that, instead of predicting 3D primitives only for the newly observed frame at each timestep and accumulating predictions over time, our method fetches current understanding of the object from the memory and refines it with the latest observation. Consequently, our approach avoids accumulation or any explicit global aggregation operations entirely, which is advantageous for object-centric reconstruction as large observation overlapping is expected over time. Thus, at each timestep, the decoded Gaussian ($\mathbf{G}_{obj,t}^{4N}$) collectively form the updated representation of the object.

**Rasterization.** During training, we employ a differentiable rasterizer to render images for photometric supervision. Specifically, at each timestep, given the predicted Gaussian representation ($\mathbf{G}_{obj,t}^{4N}$) and the ground truth poses up to time $t$, we render $t + 1$ images $\{R_{obj,t}^{i}\}_{i=0}^{t}$. In addition, we render the frame-level subsets $\mathbf{G}_{ref,t}^{N}$ and $\mathbf{G}_{src,t}^{N}$ using the poses of $V_0$ and $V_t$, respectively, producing $R_{0,t}$ and $R_{t,t}$. These additional renders are critical for training stability, as they encourage each Gaussian subgroup to specialize in reconstructing the portion of the object visible in the corresponding input view. This design, in turn, incentivizes the memory tokens to encode features that are most useful for reconstructing a complete object representation.

### 3.2 Dual-Key 3D Object Memory

To enable object-centric online reconstruction, we need a memory bank that can store the most useful features of the object and can support producing a progressively better representation of the object at each timestep. Achieving this presents two primary challenges: (1) Naive accumulation of latent features in memory will result in an ever growing memory bank, which is computationally demanding

and memory inefficient as more frames are observed. (2) Naive accumulation of predictions in the output space will result in redundant overlapping predictions, which requires additional optimization steps to consolidate and simplify the representation.

To address these limitations, we propose a novel object-centric memory mechanism, Dual-Key 3D Object Memory, that consists of a key-value memory bank. Each memory entry consists of two keys and a value, where the two keys facilitate robust readout of the most relevant memory values, via our dual-key memory readout approach. In this subsection, we first discuss how we encode the memory into the memory bank, followed by how we read from the memory. Lastly, we also introduce our memory sparsification mechanism that helps keep the memory bank compact yet effective.

**Memory Encoding** Unlike conventional single-key memories, our method leverages two complementary keys—one *latent* key and one *direction* key. Intuitively, the latent key facilitates retrieval of information to guide visual-geometrical reasoning, while the direction key provides explicit spatial guidance to improve spatial coverage during memory reading. At every timestep $t$, we encode the latent key $\mathbf{k}_t^{(L)}$ from our tokenized features using a lightweight learnable key encoder ($\mathrm{Encoder}^K$) to capture fundamental visual-geometrical cues:

$$\mathbf{k}_t^{(L)} := f_t^K = \mathrm{Encoder}^K(f_{vt}) \tag{4}$$

We further leverage a lightweight pre-trained zero-shot 3D orientation estimator [43], denoted as $\mathrm{Encoder}^D$, to encode our direction key $\mathbf{k}_t^{(D)}$. Specifically, $\mathbf{k}_t^{(D)}$ is computed based on the $\mathbb{R}^3$ axis-rotation of the object orientation $\{\theta_t, \phi_t, \gamma_t\}$:

$$\mathbf{k}_t^{(D)} := (\sin \phi_t \cos \theta_t, \ \sin \phi_t \sin \theta_t, \ \cos \phi_t) \mid \{\theta_t, \phi_t, \gamma_t\} = \mathrm{Encoder}^D(V_t') \tag{5}$$

where we use the Azimuth ($\theta_t$) and Polar ($\phi_t$) value from the prediction to convert to a unit directional vector as our directional key $\mathbf{k}_t^{(D)}$. Thus, $\mathbf{k}_t^{(L)}$ and $\mathbf{k}_t^{(D)}$ from Eq. 4 and Eq. 5 are keys used to retrieve the suitable memory values. Note that all keys and values in our object memory are at the token level, thus $\mathbf{k}_t^{(L)} \in \mathbb{R}^{p \times c}$ where $p$ is number of patches per view, we also broadcast $\mathbf{k}_t^{(D)}$ from $\mathbb{R}^{1 \times 3}$ to $\mathbb{R}^{p \times 3}$ such that patches from the same view share the same directional key. Then, after OnlineSplatter produces output tokens at time $t$, we encode the output tokens to update the value $\mathbf{v}_t^{(L)}$ corresponding to the encoded keys in Eq. 4 and Eq. 5. Specifically, a trainable value encoder (defined as $\mathrm{Encoder}^V$) takes output tokens $\mathbf{T}_{src,t}^{out}$ as input to produce the new value:

$$\mathbf{v}_t^{(L)} := f_t^V = \mathrm{Encoder}^V(\mathbf{T}_{src,t}^{out}) \tag{6}$$

Lastly, our dual-key pairs with the encoded value form a new entry at time $t$: $(\mathbf{k}_t^{(L)}, \mathbf{k}_t^{(D)}, \mathbf{v}_t^{(L)})$. The stored in-memory dual-key-values as a whole are denoted as $\mathcal{K}^{(D)} \in \mathbb{R}^{(S \cdot P) \times 3}$ and $\mathcal{K}^{(L)}, \mathcal{V}^{(L)} \in \mathbb{R}^{(S \cdot P) \times C}$ where $S$ represents the maximum size set for the object memory.

**Spatial-Guided Dual-Key Memory Reading** Our memory module maintains a large collection of tokens that encode different perspectives of the object's appearance and geometry. However, using all memory tokens for every forward pass would be computationally expensive and potentially noisy. Instead, we need an efficient mechanism to select the most relevant memory features for reconstructing the object at each timestep.

While our latent key, derived from tokenized features through end-to-end training with 3D reasoning objectives, captures both visual and geometric information, relying solely on latent key-based attention may not be optimal for object-centric reconstruction. This is because our memory readout serves two critical purposes: (1) supporting prediction for newly observed regions, and (2) retrieving the most informative features for reconstructing the complete object. To address this dual objective, we introduce a directional key that provides explicit spatial guidance for memory readout.

To perform memory read from our Object Memory $\{\mathcal{K}^{(D)}, \mathcal{K}^{(L)}, \mathcal{V}^{(L)}\}$ at timestep $t$, we treat the current (at time $t$) encoded latent key as the querying latent query (i.e. $\mathbf{q}_t^{(L)} \leftarrow \mathbf{k}_t^{(L)}$). At the same time, to compute a direction query $\mathbf{q}_t^{(D)}$, we average the directions between the current view $\mathbf{k}_t^{(D)}$ and the reference view $\mathbf{k}_0^{(D)}$, which represents an orientation that is likely to already have good coverage:

$$\mathbf{q}_t^{(D)} \leftarrow \frac{\mathbf{k}_0^{(D)} + \mathbf{k}_t^{(D)}}{\left\| \mathbf{k}_0^{(D)} + \mathbf{k}_t^{(D)} \right\|} \tag{7}$$

At each timestep, we perform two complementary memory reading operations (Orientation-Aligned Read and Orientation-Complementary Read) to retrieve memory features $f_{mem,t}$ for our OnlineSplatter transformer to reason with the reference view and current source view.

— *Orientation-Aligned Read* emphasizes memory entries that closely match both the latent and directional query keys. The similarity measure ($s_{i,t}^{(\text{align})}$) for the $i$-th memory entry is defined as:

$$s_{i,t}^{(\text{align})} = (\mathbf{q}_t^{(L)\top}\mathbf{k}_i^{(L)}) \cdot \mathbf{q}_t^{(D)\top}\mathbf{k}_i^{(D)} \cdot \tfrac{1}{\tau_t} \tag{8}$$

where $\tau_t$ is a temperature coefficient to dampen potential inaccuracies produced by $\text{Encoder}^D$. We dynamically set $\tau_t = 2.5 - \sigma_t$, where $\sigma_t \in [0, 1]$ is the confidence value of [43] for observation $V_t$.

— *Orientation-Complementary Read* retrieves memory entries that closely match the latent key but significantly differ in orientation, thus capturing complementary viewpoints. The complementary similarity score $s_i^{(\text{comp})}$ is computed as:

$$s_i^{(\text{comp})} = (\mathbf{q}_t^{(L)\top}\mathbf{k}_i^{(L)}) \cdot (-\mathbf{q}_t^{(D)\top})\mathbf{k}_i^{(D)} \cdot \tfrac{1}{\tau_t} \tag{9}$$

The final retrieved features are computed via attention-weighted sums:

$$f_{mem,t}^{(\text{align})} = \sum_{i=1}^{N} \frac{\exp(s_i^{(\text{align})})}{\sum_j \exp(s_j^{(\text{align})})}\mathbf{v}_i, \quad f_{mem,t}^{(\text{comp})} = \sum_{i=1}^{N} \frac{\exp(s_i^{(\text{comp})})}{\sum_j \exp(s_j^{(\text{comp})})}\mathbf{v}_i, \quad \text{where } i, j = 1, ..., (S \cdot P) \tag{10}$$

Overall, our dual-key memory reading approach allows the memory readout to be guided by explicit spatial cues, encouraging spatial coverage on top of the visual-geometrical cues.

**Memory Sparsification Mechanism** Furthermore, we would like to maintain a bounded memory size while preserving good coverage of diverse viewpoints. To achieve this, inspired by attention-based memory design in [3, 39], we propose a memory sparsification strategy that leverages our dual key design. Specifically, when the memory reaches its max capacity $S$, we drop $20\%$ of memory that is deemed least useful by considering two factors: (i) usage (cross-attention contribution), and (ii) spatial coverage (average angular distance to other entries). Specifically, for each memory entry $i$, we define the total usage $U_i$ by averaging the accumulated cross-attention weight:

$$U_i = \frac{1}{|\mathcal{T}_i|} \sum_{t \in \mathcal{T}_i} \Big[ \mathcal{A}_{i,t}^{(\text{align})} + \mathcal{A}_{i,t}^{(\text{comp})} \Big], \tag{11}$$

where $\mathcal{T}_i$ is the set of timesteps in which entry $i$ participated, and $\mathcal{A}_{i,t}^{(\cdot)} = \frac{\exp(s_i^{(\cdot)})}{\sum_j \exp(s_j^{(\cdot)})}$, i.e., $\mathcal{A}_{i,t}^{(\cdot)}$ denotes the normalized cross-attention weight derived from Eq. (10) for memory reads. Intuitively, $U_i$ is larger if an entry has been repeatedly or strongly attended to, meaning that it is useful. Subsequently, we compute spatial coverage for each memory entry. Let $\mathbf{k}_i^{(D)} \in \mathbb{R}^3$ be the direction key for entry $i$. We define its coverage measure $C_i$ as the *average* angular dot product to all other entries:

$$C_i = \frac{1}{N - 1} \sum_{\substack{j=1 \\ j \neq i}}^{N} \mathbf{k}_i^{(D)\top}\mathbf{k}_j^{(D)}, \tag{12}$$

Lastly, to prune object memory, we sort all entries by $C_i$ and select the top $50\%$ of entries as the *dense subset*, which represent viewpoints in memory that are spatially well covered. From this dense subset, we remove the $40\%$ lowest-ranked entries with respect to usage $U_i$. Formally,

$$\text{PruneSubset} = \Big\{ i \in \text{DenseSubset} \ \Big| \ U_i \leq U_q \Big\}, \tag{13}$$

where $U_q$ is the usage value at the 40-th percentile within $\text{DenseSubset}$. These pruned entries each time collectively constitute $20\%$ of the full memory cap, balancing between retaining unique coverage and discarding underused features.

## 3.3 Training and Inference

**Staged Training.** Our OnlineSplatter model is trained to optimize two complementary objectives: (1) learning relative object-camera pose relationships and predicting pixel-aligned 3D Gaussian

parameters, and (2) effectively utilizing our attention-based Object Memory to reason about the object in canonical coordinate space through memory encoding and reading. These objectives present a challenging optimization landscape, as the gradients for the second objective only become meaningful after the first objective reaches a certain level of convergence. To address this, we employ a two-stage training strategy: **(1) Warm-up Training:** We train the core reconstruction components without the Object Memory module. Specifically, we optimize the view encoder ($\text{Encoder}_1^I$), positional and view embeddings ($f_{pos}^{emb}$ and $f_{view}^{emb}$), OnlineSplatter transformer, and unpatchify decoder in the first stage. **(2) Main Training:** We include the Object Memory module and train the entire network end-to-end, allowing the model to learn both reconstruction and memory simultaneously.

**Training Loss.** We employ a combination of both photometric and geometrical losses to train our model. First, the photometric loss $\mathcal{L}_{\text{photo}}$ minimizes the MSE between the ground truth images and rendered images from predicted 3D Gaussian parameters at ground truth camera poses, where $\mathcal{L}_{\text{photo}}$ is computed on object regions only. We include a background penalty term ($\mathcal{L}_{\text{bg}}$) in $\mathcal{L}_{\text{photo}}$ to penalize Gaussians' color and opacity outside the object's visual hull. While the photometric loss alone can theoretically supervise feedforward reconstruction tasks [51], geometrical supervision can often enhance convergence speed and reconstruction quality [18, 2, 14, 29]. Thus, we incorporate geometrical loss $\mathcal{L}_{\text{geo}}$, which includes a ray alignment component ($\mathcal{L}_{\text{ray}}$) to ensure that predicted 3D points lie along their corresponding camera rays, and a depth component ($\mathcal{L}_{\text{depth}}$) to minimize MSE between predicted and ground truth relative depths. The **overall training objective** is: $\mathcal{L}_{\text{total}} = \mathcal{L}_{\text{photo}} + \lambda_g \mathcal{L}_{\text{geo}}$, where $\lambda_g$ balances the contribution of geometrical supervision. We provide full details in the appendix.

**Implementation Details.** We train and evaluate our model on $256 \times 256$ resolution images. We use 8x A100 GPUs for $250K$ steps with a batch size of $64$ in the *Warm-up Training* stage and $500K$ steps with a batch size of $16$ in the *Main Training* stage. We sample 3-5 views per object during *warm-up* and 6-12 views per object during *main* training. We provide more details on implementation and hyperparameters in the appendix.

# 4 Experiments

This section evaluates our approach by outlining the evaluation protocol, describing the datasets for training and testing, comparing against state-of-the-art baselines, and conducting ablation studies to analyze each component's impact.

## 4.1 Experimental settings

**Evaluation Protocol** To properly evaluate our online object reconstruction framework, we need to assess how well it performs at different stages of observation accumulation. This is crucial because real-world applications often require reliable reconstruction even with limited initial observations. We therefore design a stage-wise evaluation protocol that examines performance across three distinct phases: **1) Early Stage** ($\mathcal{T}_{\text{early}} := \{1 \leq t \leq 4\}$): Tests the model's ability to quickly establish an initial object representation with minimal observations; **2) Mid Stage** ($\mathcal{T}_{\text{mid}} := \{5 \leq t \leq 10\}$): Evaluates how well the model refines its reconstruction as more views become available; **3) Late Stage** ($\mathcal{T}_{\text{late}} := \{11 \leq t \leq T\}$): Assesses the model's capability to maintain and improve reconstruction quality with extended observation sequences. For each test sequence of $N$ frames $\{V_n\}_{n=1}^N$, we split the frames into two sets: **Input frames** ($\mathcal{V}_{\text{input}}$): A randomly sampled subset of $\frac{N}{2}$ frames used for input; **Target frames** ($\mathcal{V}_{\text{target}}$): The remaining $\frac{N}{2}$ frames reserved for NVS-based evaluation. Specifically, during evaluation, each frame $V_t$ is provided to the model at each timestep $t \in \{1, \ldots, T\}$ (where $T = \frac{N}{2}$) and we expect the model to output a 3D object representation $\hat{O}_t$ at each timestep $t$ and produce novel view images $\hat{R}_{v,t}$ for all target viewpoints $v \in \mathcal{V}_{\text{target}}$. We then compare these renders against the corresponding ground-truth frames $V_v$. The stage-wise performance is then computed as: $\mathcal{M}_{\text{stage}} = \frac{1}{|\mathcal{T}_{\text{stage}}|} \sum_{t \in \mathcal{T}_{\text{stage}}} \sum_{v \in \mathcal{V}_{\text{target}}} \mathcal{M}(\hat{R}_{v,t}, V_v)$ where $\mathcal{M}$ represents standard image quality metrics used for 3D reconstruction [16, 51, 2]: PSNR, SSIM, and LPIPS.

**Datasets.** We train on 100K objects sampled from Objaverse [5, 4]. Unlike conventional few-view or image-to-3D setups that render from biased polar angles and fixed upright poses, our setting targets real-world freely moving object reconstruction, where objects appear in arbitrary poses relative to the camera. This difference is critical, as real-world data often includes partial views with unknown

| | GSO | | | | | | | | |
|---|---|---|---|---|---|---|---|---|---|
| **Method** | **Early-Stage** | | | **Mid-Stage** | | | **Late-Stage** | | |
| | PSNR↑ | SSIM↑ | LPIPS↓ | PSNR↑ | SSIM↑ | LPIPS↓ | PSNR↑ | SSIM↑ | LPIPS↓ |
| $FSO_{rand4}$ | 21.358 | 0.861 | 0.177 | 21.919 | 0.877 | 0.181 | 21.737 | 0.855 | 0.181 |
| $FSO_{dist4}$ | 22.365 | 0.874 | 0.119 | 23.757 | 0.862 | 0.117 | 23.751 | 0.873 | 0.120 |
| $NPS_{dist2}$ | 22.986 | 0.859 | 0.155 | 23.050 | 0.863 | 0.162 | 22.949 | 0.878 | 0.156 |
| $NPS_{dist3}$ | 23.331 | 0.862 | 0.149 | 23.206 | 0.861 | 0.138 | 24.141 | 0.863 | 0.125 |
| **Ours** | **26.329** | **0.921** | **0.084** | **27.553** | **0.933** | **0.066** | **31.737** | **0.969** | **0.075** |

| | HO3D | | | | | | | | |
|---|---|---|---|---|---|---|---|---|---|
| **Method** | **Early-Stage** | | | **Mid-Stage** | | | **Late-Stage** | | |
| | PSNR↑ | SSIM↑ | LPIPS↓ | PSNR↑ | SSIM↑ | LPIPS↓ | PSNR↑ | SSIM↑ | LPIPS↓ |
| $FSO_{rand4}$ | 18.488 | 0.820 | 0.187 | 18.552 | 0.817 | 0.191 | 17.683 | 0.810 | 0.199 |
| $FSO_{dist4}$ | 18.594 | 0.837 | 0.177 | 19.215 | 0.848 | 0.184 | 19.619 | 0.843 | 0.183 |
| $NPS_{dist2}$ | 21.063 | 0.855 | 0.160 | 22.803 | 0.841 | 0.158 | 22.134 | 0.846 | 0.164 |
| $NPS_{dist3}$ | 21.134 | 0.853 | 0.162 | 22.967 | 0.869 | 0.165 | 22.947 | 0.860 | 0.163 |
| **Ours** | **23.627** | **0.910** | **0.152** | **25.803** | **0.912** | **0.122** | **27.928** | **0.952** | **0.099** |

Table 1: Comparison of different baselines on two datasets. Results are shown for early-stage, mid-stage, and late-stage settings. Best results are **bolded** and second best results are underlined.

object centers, rendering naive pre-processing ineffective. To simulate such conditions, we develop a custom script (details in the appendix) that generates diverse trajectories with look-at jitter, varying focal lengths, and randomized lighting. Each object receives a unique trajectory, ensuring diverse motion patterns for training.

For evaluation, we use two datasets of unseen objects. First, we test on Google Scanned Objects (GSO) [7], rendering 36 frames per object using our training pipeline (each with distinct lighting and motion). Second, we assess generalization to real-world monocular videos with occlusions using the HO3D dataset [10], which contains hand-object interaction sequences.

**Baselines.** No prior feed-forward model supports pose-free, RGB-only reconstruction in online settings. We thus adapt two leading pose-free few-view methods, FreeSplatter [48] and NoPoSplat [51], to our online setting. Both consume uncalibrated RGB images and predict 3D Gaussians.

*FreeSplatter-O* [48] is a transformer-based, object-centric method trained on Objaverse. It jointly processes 4 pose-free views per sample. To adapt it online, we introduce two frame selection strategies for each timestep: (1) *rand4*: randomly selects 4 frames from past observations ($FSO_{rand4}$); (2) *dist4*: selects 4 frames with the largest feature differences using a DINO [1] encoder ($FSO_{dist4}$).

*NoPoSplat* [51] extends DUSt3R [42] backbone with 3D Gaussian output heads and is fine-tuned on scene-level data with 2-3 input views. We adapt it for object-centric reconstruction by fine-tuning on Objaverse with object mask supervision, and apply the *dist* strategy to select 2 or 3 diverse frames, resulting in $NPS_{dist2}$ and $NPS_{dist3}$.

## 4.2 Results

Table 1 compares our method against strong baselines on both GSO and HO3D datasets across different stages of observation accumulation. GSO offers statistical significance through its diverse set of unseen objects, while HO3D introduces real-world challenges, including complex interactions and occlusions. Across all metrics and stages, OnlineSplatter achieves superior performance—improving up to +7.596 PSNR and +0.106 SSIM on GSO, and +4.981 PSNR and +0.092 SSIM on HO3D in late-stage reconstruction. Two key insights emerge: good early-stage performance and temporal development. Even with fewer than four observations, OnlineSplatter significantly outperforms all baselines. Over time, a clear divergence appears. Baselines using explicit frame selection often exhibit unstable or stagnant performance. In contrast, OnlineSplatter consistently improves with more observations, as also shown qualitatively in Fig. 3, where our method delivers notably better visual quality and geometric accuracy from early to late stages. This underscores the strength of our Object Memory mechanism in leveraging temporal cues for progressive reconstruction refinement.

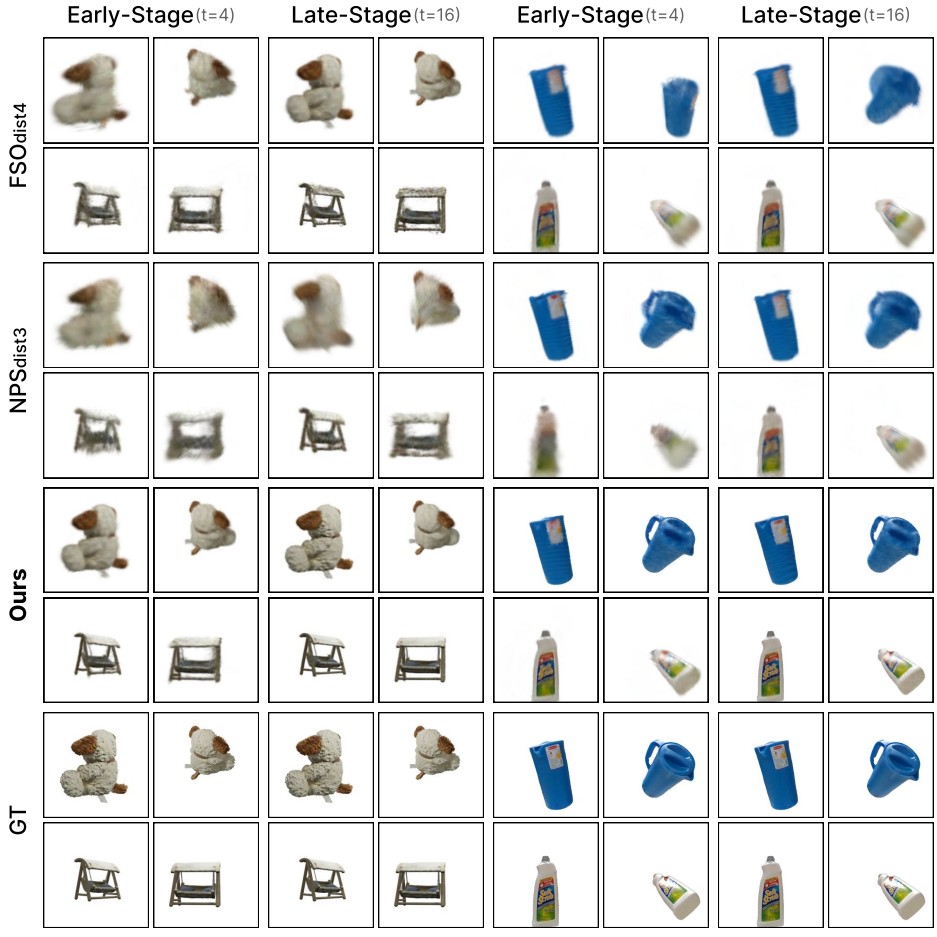

Figure 3: Qualitative results of different baselines and our method on the GSO (left) and HO3D (right) datasets. We visualize the results at inference timestep $t = 4$ and $t = 16$, which corresponds to the early-stage and late-stage settings, respectively. Our reconstructed outputs show significantly better visual quality and geometric accuracy as more observations become available.

| Variants | Early-Stage $\mathcal{M}_{avg} \uparrow$ | Mid-Stage $\mathcal{M}_{avg} \uparrow$ | Late-Stage $\mathcal{M}_{avg} \uparrow$ |
|---|---|---|---|
| **Ours** | 0.699 | 0.734 | 0.810 |
| w/o latent key | 0.545 | 0.582 | 0.596 |
| w/o direction key | 0.699 | 0.701 | 0.723 |
| w/ encode from gs | 0.541 | 0.582 | 0.611 |
| w/ random pruning | 0.697 | 0.728 | 0.764 |

Table 2: Impact of dual-key object memory design. Results are reported on GSO dataset.

| Variants | Early-Stage $\mathcal{M}_{avg} \uparrow$ | Mid-Stage $\mathcal{M}_{avg} \uparrow$ | Late-Stage $\mathcal{M}_{avg} \uparrow$ |
|---|---|---|---|
| **Ours** | 0.699 | 0.734 | 0.810 |
| w/o staged training | 0.545 | 0.582 | 0.588 |
| w/o ray loss ($\mathcal{L}_{ray}$) | 0.562 | 0.599 | 0.682 |
| w/o bg penalty ($\mathcal{L}_{bg}$) | 0.675 | 0.712 | 0.795 |
| w/o depth loss ($\mathcal{L}_{depth}$) | 0.691 | 0.728 | 0.805 |
| w/ sequential sampling only | 0.645 | 0.682 | 0.688 |
| w/ random sampling only | 0.697 | 0.728 | 0.764 |

Table 3: Impact of training strategy components. Results are reported on GSO dataset.

## 4.3 Ablations and Analysis

In this section, we ablate different components of our method and analyze the results. For better readability, we normalize and average metrics of PSNR, SSIM, and LPIPS to $[0, 1]$ and report the $\mathcal{M}_{avg}$ value, where a higher value indicates better performance, more details in the appendix.

**Impact of Object Memory Design.** We conduct ablation studies to validate our dual-key object memory design in Sec. 3.2, summarizing results on the GSO dataset in Table 2. Specifically: *Dual-key Design:* Removing the latent key severely degrades performance at all stages due to loss of visual-geometrical cues. Besides, removing the direction key significantly impacts later stages, highlighting its role in spatial coverage. *Memory Encoding:* Encoding memory from unpatchified

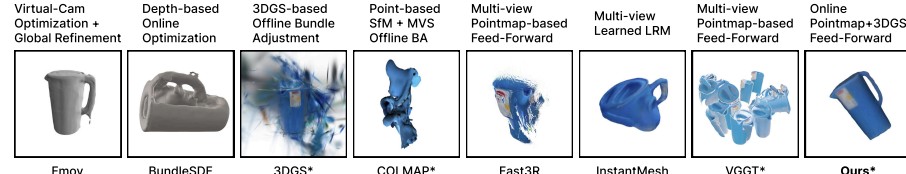

Figure 4: Visual comparison of mesh results between different methods. Methods marked with an asterisk (*) indicate that additional pre- or post-processing steps were applied to generate the visual results. More details in the appendix.

Gaussian parameters rather than direct Transformer token outputs leads to degraded performance, showing that using Transformer tokens for memory encoding better preserves learned representations. *Memory Sparsification:* Our sparsification strategy based on usage rate and spatial coverage outperforms random pruning, particularly in later stages, showing the efficacy of our pruning criteria.

**Impact of Training Strategy.** We evaluate our training strategy (discussed in Sec. 3.3) through ablation studies and show results in Table 3, demonstrating: *Staged Training:* Removing the two-stage training (warm-up followed by main training) by using a single stage only greatly lowers performance, validating our intuition that the memory module requires a well-initialized backbone. *Loss Components:* Removing the ray alignment ($\mathcal{L}_{\mathrm{ray}}$) notably reduces convergence speed and stability, harming performance. Excluding the visual hull ($\mathcal{L}_{\mathrm{bg}}$) penalty moderately degrades performance, underscoring its role in preserving object boundaries. Removing depth term ($\mathcal{L}_{\mathrm{depth}}$) slightly impacts performance, as it primarily aids convergence during warm-up training. *Training Frame Sampling:* Our progressive sampling strategy (discussed in the appendix) outperforms sequential-only and random-only alternatives, underscoring the advantage of curriculum learning in our training.

**Mesh Visual Comparison.** To demonstrate our approach's efficacy, we convert our final 3DGS representation into meshes and visually compare it comprehensively with state-of-the-art methods from different paradigms. We evaluate against representative methods: (1) those leveraging ground-truth depth information during optimization [44], (2) offline bundle adjustment techniques performing global optimization across frames [31, 16, 30], (3) diffusion-based approaches using learned 3D priors [47], and (4) recent feed-forward pointmap models operating offline [49, 40]. As shown in Fig. 4, despite being provided with object masks, many methods struggle to reconstruct freely moving objects. In contrast, our method achieves comparable quality to those requiring extensive optimization or additional depth supervision while retaining the benefits of a feed-forward, online framework.

## 5 Limitations and Future Work

Our current framework has some limitations that warrant attention. First, the framework outputs 3D Gaussian Splatting (3DGS) representations, which while efficient for object understanding and rendering, may not be directly suitable for certain downstream applications requiring explicit mesh representations. Converting 3DGS to meshes robustly remains challenging and is an active area of research. Future work could explore hybrid representations that maintain both rendering efficiency and mesh compatibility. Second, the framework's performance may depend on the quality of the initial reference view. Poor initial observation, such as heavily occluded or blurry first frames, could impact subsequent reconstruction quality. Lastly, our framework is currently limited to rigid objects. Future work could explore modeling non-rigid objects and integrate it with downstream tasks like robotic manipulation.

## 6 Conclusion

We presented OnlineSplatter, a novel framework for pose-free online 3D reconstruction of freely moving objects from monocular RGB video. Our dual-key memory module enables efficient temporal feature fusion with constant computational complexity. Extensive evaluation demonstrates superior reconstruction quality without requiring camera poses, depth priors, or global optimization. This makes our approach particularly suitable for robotics applications in dynamic environments requiring online continuous moving object perception and manipulation.

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

# A    Notations and Definitions

Please refer to Table 4 for the list of symbols and their definitions used in the paper.

# B    Additional Details on Training Losses

In Section 3.3 of the main paper, we introduced our losses used for training. Below, we give more details.

**Photometric Supervision** For photometric supervision, we employ a differentiable rasterizer to render 2D images from the predicted 3D Gaussian parameters ($\mathbf{G}_{obj,t}^{4N}$) and ground truth camera poses. The photometric loss $\mathcal{L}_{\text{photo}}$ consists of two components: $\mathcal{L}_{\text{photo}} = \mathcal{L}_{\text{masked}} + \lambda_{\text{bg}} \mathcal{L}_{\text{bg}}$, where $\mathcal{L}_{\text{masked}}$ is the masked MSE loss computed only on object regions:

$$\mathcal{L}_{\text{masked}} = \frac{1}{|\mathcal{S}_t|} \sum_{p \in \mathcal{S}_t} \| R_t(p) - V_t(p) \|_2^2 \tag{14}$$

Here, $\mathcal{S}_t$ denotes the set of pixels belonging to the object silhouette at time $t$, $R_t(p)$ and $V_t(p)$ respectively refer to the rendered color and ground truth color at pixel $p$ at time $t$. The background penalty term $\mathcal{L}_{\text{bg}}$ penalizes Gaussians's color and opacity outside the object's visual hull:

$$\mathcal{L}_{\text{bg}} = \frac{1}{|\mathcal{G}_t|} \sum_{g \in \mathcal{G}_t} (\|\mathbf{c}_g\|_2^2 + \alpha o_g) \tag{15}$$

where $\mathcal{G}_t \subset \mathbf{G}_{obj,t}^{4N}$ is the subset of predicted Gaussians that is outside of the object's visual hull defined by object mask from reference view ($M_0$) and current view ($M_t$), $o_g$ is the opacity of Gaussian $g$, and $\alpha$ is a weighting factor.

**Geometrical Supervision** While the photometric loss alone can theoretically supervise feedforward reconstruction tasks [51], we incorporate geometrical supervision to enhance convergence speed and reconstruction quality. This is motivated by prior work [18, 2, 14, 29] showing that Gaussian means significantly impact 3DGS convergence. To handle potential missing or noisy depth ground truth, we decompose the geometrical loss into two terms: $\mathcal{L}_{\text{geo}} = \mathcal{L}_{\text{ray}} + \lambda_d \mathcal{L}_{\text{depth}}$. The ray alignment loss $\mathcal{L}_{\text{ray}}$ ensures that predicted 3D points lie along their corresponding camera rays:

$$\mathcal{L}_{\text{ray}} = \frac{1}{|\mathcal{S}_t|} \sum_{p \in \mathcal{S}_t} (1 - r_p \cdot \hat{r}_p) \tag{16}$$

where $r_p$ is the normalized predicted ray from camera center to pixel $p$. The normalized depth loss $\mathcal{L}_{\text{depth}}$ compares relative depths:

$$\mathcal{L}_{\text{depth}} = \frac{1}{|\mathcal{S}_t|} \sum_{p \in \mathcal{S}_t} \| \frac{d_p}{\bar{d}} - \frac{z_p}{\bar{z}} \|_2^2 \tag{17}$$

where $d_p$ and $z_p$ are predicted and ground truth depths, and $\bar{d}$ and $\bar{z}$ are their respective means.

**Overall training objectives.** The final training objective combines both photometric and geometrical losses: $\mathcal{L}_{\text{total}} = \mathcal{L}_{\text{photo}} + \lambda_g \mathcal{L}_{\text{geo}}$ where $\lambda_g$ balances the contribution of geometrical supervision.

# C    Additional Analysis

## C.1    Impact of Training Data Scaling and Selection.

While the complete Objaverse [5, 4] dataset contains over $10M$ objects, we conduct a systematic study of our method's scalability under computational constraints by examining two critical factors: (1) the relationship between dataset size and model performance, scaling from $1K$ to $100K$ objects, and (2) the impact of data curation strategies on reconstruction quality. To evaluate these factors, we compare two sampling approaches: random selection from the full Objaverse dataset versus quality-based selection using aesthetic scores from [46]. All experiments are conducted with identical training configurations—500K training steps and consistent hyperparameters—to ensure a fair and controlled comparison.

| Section | Symbol | Definition |
|---|---|---|
| Input and Output | $V_t$ | RGB input frame at timestep $t$ |
| | $M_t$ | Object mask at timestep $t$ for frame $V_t$ |
| | $V'_t$ | Masked RGB frame at timestep $t$ with background removed |
| | $\hat{O}_t$ | Predicted 3D object representation at timestep $t$ |
| | $\hat{R}_{v,t}$ | Rendered novel view image for viewpoint $v$ at timestep $t$ |
| Token Representations | $\mathbf{T}^{in}_{ref}$ | Reference-view input tokens (from initial frame) |
| | $\mathbf{T}^{in}_{src,t}$ | Source-view input tokens at timestep $t$ |
| | $\mathbf{T}^{in}_{mem,t}$ | Memory-readout input tokens at timestep $t$ |
| | $\mathbf{T}^{in}_{view}$ | Generic input tokens for view $\in \{ref, src, mem\}$ |
| | $\mathbf{T}^{out}_{ref}$ | Reference-view output tokens |
| | $\mathbf{T}^{out}_{src,t}$ | Source-view output tokens at timestep $t$ |
| | $\mathbf{T}^{out}_{mem,t}$ | Memory-readout output tokens at timestep $t$ |
| | $f_{view}$ | Encoded features for a view $\in \{ref, src, mem\}$ |
| | $f^{emb}_{pos}$ | Positional embeddings |
| | $f^{emb}_{view}$ | View-specific embeddings for view $\in \{ref, src, mem\}$ |
| Gaussian Parameters | $\mathbf{G}^{4N}_{obj,t}$ | Complete object Gaussian representation at timestep $t$ |
| | $\mathbf{G}^{2N}_{mem,t}$ | Memory-based subset of Gaussians at timestep $t$ |
| | $\mathbf{G}^{N}_{ref,t}$ | Reference-view subset of Gaussians at timestep $t$ |
| | $\mathbf{G}^{N}_{src,t}$ | Source-view subset of Gaussians at timestep $t$ |
| | $\mathbf{G}^{K}$ | Set of $K$ Gaussian primitives |
| | $\boldsymbol{\mu}_k$ | Mean position of $k$-th Gaussian |
| | $\mathbf{r}_k$ | Rotation of $k$-th Gaussian |
| | $\mathbf{s}_k$ | Scale of $k$-th Gaussian |
| | $\mathbf{c}_k$ | Color / Spherical harmonics coefficients of $k$-th Gaussian |
| | $o_k$ | Opacity of $k$-th Gaussian |
| Memory Module | $\mathbf{k}^{(L)}_t$ | Latent key at timestep $t$ |
| | $\mathbf{k}^{(D)}_t$ | Direction key at timestep $t$ |
| | $\mathbf{v}^{(L)}_t$ | Memory value at timestep $t$ |
| | $\mathcal{K}^{(D)}$ | Set of stored directional keys |
| | $\mathcal{K}^{(L)}$ | Set of stored latent keys |
| | $\mathcal{V}^{(L)}$ | Set of stored memory values |
| | $\mathbf{q}^{(L)}_t$ | Latent query at timestep $t$ |
| | $\mathbf{q}^{(D)}_t$ | Direction query at timestep $t$ |
| | $s^{(align)}_{i,t}$ | Alignment similarity score for entry $i$ at timestep $t$ |
| | $s^{(comp)}_i$ | Complementary similarity score for entry $i$ |
| | $\tau_t$ | Temperature coefficient at timestep $t$ |
| | $\sigma_t$ | Confidence value from orientation estimator |
| | $U_i$ | Usage measure for memory entry $i$ |
| | $C_i$ | Coverage measure for memory entry $i$ |
| | $\mathcal{T}_i$ | Set of timesteps for memory entry $i$ |
| | $\mathcal{A}^{(\cdot)}_{i,t}$ | Normalized cross-attention weight |
| Evaluation Metrics | $\mathcal{M}$ | Generic image quality metric (PSNR, SSIM, LPIPS) |
| | $\mathcal{M}_{\text{stage}}$ | Stage-wise performance metric |
| | $\mathcal{M}_{avg}$ | Normalized average of metrics |
| | $\mathcal{V}_{\text{input}}$ | Set of input frames |
| | $\mathcal{V}_{\text{target}}$ | Set of target frames |
| | $\mathcal{T}_{\text{stage}}$ | Set of timesteps in a stage |
| Dimensions and Constants | $N$ | Number of pixels per frame ($H \times W$) |
| | $S$ | Maximum memory size |
| | $P$ | Number of patches per view |
| | $C$ | Feature dimension |

Table 4: List of mathematical notations used throughout the paper.

As shown in Fig. 5, our analysis reveals two key findings. First, our method demonstrates strong scaling behavior, with performance continuing to improve as the dataset size increases to $100K$ objects without showing signs of saturation. This suggests significant potential for further gains with larger datasets. Second, we find that careful data curation through aesthetic quality filtering yields substantial performance improvements, indicating that strategic data selection can be an effective approach for optimizing model performance under limited computational resources.

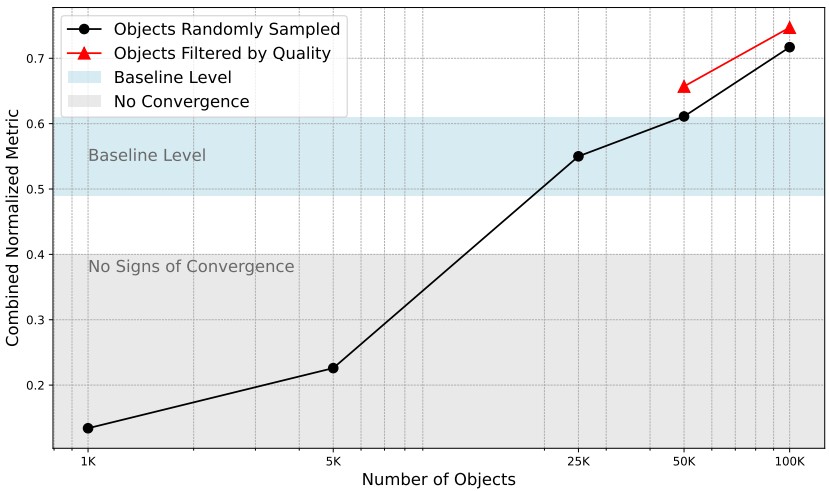

Figure 5: Impact of Training Data Quantity and Quality.

## C.2   Impact of Ray Alignment Loss in Geometrical Supervision.

While photometric RGB-based loss can effectively supervise 3D Gaussian positions when they are already well-aligned with the ground truth and visible from the rendered pose, it struggles to guide Gaussians that are far from their optimal positions. This limitation becomes particularly problematic in object-centric reconstruction, where a significant portion of the region in observed frame consists of plain background regions that do not receive meaningful supervision. In such cases, predicted Gaussians may either float arbitrarily in space or be suppressed by the background penalty term ($\mathcal{L}_{\mathrm{bg}}$), failing to contribute meaningfully to object reconstruction.

To address this challenge and accelerate convergence, we introduce a ray alignment term $\mathcal{L}_{\mathrm{ray}}$ that explicitly regularizes the 3D Gaussian positions (as detailed in Sec. B). This term ensures that predicted Gaussians align with their corresponding camera rays in the object region, providing crucial geometric guidance even when photometric supervision is insufficient.

To demonstrate the effectiveness of the ray alignment loss, we conduct a comparative analysis between training with and without $\mathcal{L}_{\mathrm{ray}}$, visualized in Fig. 6. The visualization shows two camera views from a training sample at different stages (1K-10K training steps). In each view, blue lines represent ground-truth per-pixel camera rays, while red lines indicate predicted 3D rays from the ground-truth camera center to the predicted Gaussian means. The comparison reveals two key findings:

- Without $\mathcal{L}_{\mathrm{ray}}$, many Gaussians drift away from the object region, effectively becoming "dead" primitives that contribute little to reconstruction.

- With $\mathcal{L}_{\mathrm{ray}}$, Gaussians maintain better alignment with ground-truth rays in object regions, and background Gaussians actively migrate toward object regions to participate in reconstruction.

These observations confirm that the ray alignment loss serves as an effective regularizer for 3D Gaussian positions and significantly improves convergence speed during training. The quantitative results also reflect this significant improvement (See Table 5).

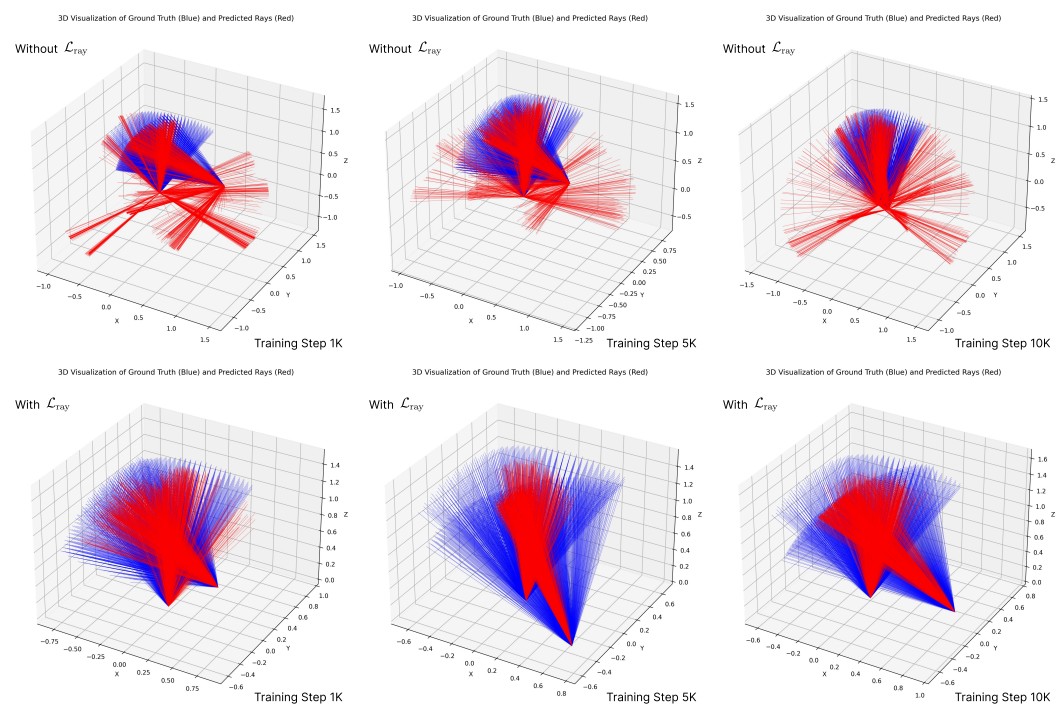

Figure 6: Visualization of the effect of without (top row) and with (bottom row) ray alignment loss $\mathcal{L}_{\mathrm{ray}}$ over $1K - 10K$ training steps. The visualization shows both the ground-truth per-pixel camera rays (in blue) and the predicted 3D ray pointing from the ground-truth camera center to the predicted 3D point (in red). The qualitative visualization shows clearly that the ray alignment loss is effective in regularizing 3D gaussians positions and converges quickly.

## C.3   Additional Notes on Ablation Study.

In the main paper, we use the averaged metrics ($\mathcal{M}_{avg}$) of PSNR, SSIM, and LPIPS to report results in ablation studies (Table 2 and Table 3) due to space limitation and better readability. Note that $\mathcal{M}_{avg}$ is computed as $\mathcal{M}_{avg} = \frac{1}{3}\Big[\mathrm{clip}\big(\frac{\mathrm{PSNR}-20}{20},\ 0,\ 1\big) + \mathrm{SSIM} + \Big(1 - \mathrm{clip}\big(\frac{\mathrm{LPIPS}}{0.6},\ 0,\ 1\big)\Big)\Big]$ where we normalize PSNR, SSIM, and LPIPS to [0, 1] by clipping and scaling such that a higher $\mathcal{M}_{avg}$ value indicates relatively better performance. Here in Table 5, we provide the detailed original results for completeness.

| Impact of Dual-key Object Memory Design | | | | | | | | |
|---|---|---|---|---|---|---|---|---|
| **Method** | **Early-Stage** | | | **Mid-Stage** | | | **Late-Stage** | | |
| | PSNR↑ | SSIM↑ | LPIPS↓ | PSNR↑ | SSIM↑ | LPIPS↓ | PSNR↑ | SSIM↑ | LPIPS↓ |
| **Ours** | 26.329 | 0.921 | 0.084 | 27.553 | 0.933 | 0.066 | 31.737 | 0.969 | 0.075 |
| w/o latent key | 23.260 | 0.764 | 0.176 | 23.947 | 0.805 | 0.153 | 24.212 | 0.822 | 0.147 |
| w/o direction key | 26.320 | 0.917 | 0.084 | 26.385 | 0.923 | 0.083 | 31.147 | 0.749 | 0.083 |
| w encode from gs | 23.237 | 0.759 | 0.179 | 23.983 | 0.804 | 0.154 | 24.538 | 0.837 | 0.138 |
| w/ random pruning | 26.261 | 0.919 | 0.086 | 27.089 | 0.927 | 0.080 | 31.443 | 0.852 | 0.080 |
| Impact of Training Strategy Components | | | | | | | | |
| **Method** | **Early-Stage** | | | **Mid-Stage** | | | **Late-Stage** | | |
| | PSNR↑ | SSIM↑ | LPIPS↓ | PSNR↑ | SSIM↑ | LPIPS↓ | PSNR↑ | SSIM↑ | LPIPS↓ |
| **Ours** | 26.329 | 0.921 | 0.084 | 27.553 | 0.933 | 0.066 | 31.737 | 0.969 | 0.075 |
| w/o staged training | 23.257 | 0.766 | 0.177 | 24.016 | 0.803 | 0.155 | 24.169 | 0.808 | 0.151 |
| w/o ray loss | 23.528 | 0.788 | 0.166 | 24.312 | 0.823 | 0.145 | 25.962 | 0.905 | 0.094 |
| w/o bg penalty | 25.844 | 0.896 | 0.097 | 26.562 | 0.937 | 0.077 | 31.714 | 0.949 | 0.091 |
| w/o depth loss | 26.147 | 0.917 | 0.091 | 28.301 | 0.921 | 0.150 | 31.643 | 0.960 | 0.077 |
| w/ sequential sampling only | 25.268 | 0.866 | 0.117 | 25.929 | 0.904 | 0.093 | 26.182 | 0.905 | 0.091 |
| w/ random sampling only | 26.249 | 0.920 | 0.085 | 27.587 | 0.923 | 0.097 | 31.695 | 0.906 | 0.119 |

Table 5: Expanded version of Table 2 and Table 3 in the Main Paper

# D  Additional Training and Implementation Details

In Section 3.3 of the main paper, due to space constraints, we provided only the most significant training and implementation details. Here, we provide comprehensive details on our training (Section D.1 and D.2) and implementation (Section D.3).

## D.1  Progressive Training Frame Sampling.

Most methods that leverage monocular video data rely on implicit temporal continuity—i.e., minimal changes between consecutive frames—during training [39, 9, 51], which substantially simplifies the learning problem. In contrast, our setting involves dynamic motion from both the camera and the target object, with no assumptions made about the nature of the interacting agent (e.g., human hand or robotic manipulator). As a result, the relative object-camera pose, especially the angular velocity, may vary significantly between adjacent frames. To enable the model to learn across a wide range of motion dynamics, we leverage a **curriculum-based progressive frame sampling strategy**. Training begins with temporally-close frames, gradually increasing the frame interval (thereby reducing co-visibility), and eventually transitions to fully random frame sampling. This staged approach ensures exposure to diverse object motions throughout training. As a result, the trained model generalizes well to both sequential video data and unordered image sets.

## D.2  Training Data Rendering Pipeline.

For each object sampled from the Objaverse dataset, we generate a unique fly-around sequence using a *custom Blender script* that introduces controlled randomization across camera motion, optical parameters, and scene illumination. Our pipeline operates as follows:

First, we construct a smooth camera trajectory by sampling $K_1=4$ elevation angles $\{\theta_k\}$ and $K_2=8$ radius values $\{r_k\}$ within a spherical shell $[d_{\min}, d_{\max}]$. This sampling strategy ensures comprehensive coverage of both polar and azimuthal viewing angles. Through linear interpolation of these key points combined with uniformly distributed azimuth angles $\phi$, we generate $N=100$ waypoints that exhibit smooth transitions in both vertical and radial dimensions. We then construct a periodic cubic spline through these waypoints and uniformly sample it to obtain the final camera positions. By re-seeding the random number generator for each object, we ensure that every sequence features a unique trajectory with object-specific zoom patterns.

During rendering, we employ a `Track-To` constraint to maintain camera focus on the object while introducing a small per-frame *look-at jitter* ($\pm5$ cm on each axis) to prevent the object from remaining perfectly centered. To further enhance diversity, we randomly sample focal lengths from $\{30, 35, 40, 45, 50\}$ mm and configure a combination of area and point lights with randomized positions and intensities drawn from broad uniform distributions. This comprehensive approach yields training sequences characterized by diverse motion patterns, controlled scale variations, and rich illumination conditions.

## D.3  Implementation Details.

We provide comprehensive implementation details of our OnlineSplatter framework below.

**Model Architecture.** Our model consists of several key components:

- **Image Encoders:** We use a frozen DINO backbone [1] as $\text{Encoder}_1^I$ with patch size 8 and output dimension 768. The learnable $\text{Encoder}_2^I$ follows the same architecture but is output dimension is 256. The concatenation of $\text{Encoder}_1^I$ and $\text{Encoder}_2^I$ provides a 1024 dimensional feature vector for each patch token.

- **OnlineSplatter Transformer:** The transformer processes tokenized inputs through 24 layers, each with 16 attention heads and hidden dimension 1024. We use layer normalization and a dropout rate of 0.05.

- **Memory Module:** The object memory maintains a maximum of $S = 1024 \times 20$ entries, with each entry containing token-level features of dimension $C = 1024$. The number of patches per view $P = 1024$ is determined by the input resolution $256 \times 256$ and patch size 8. The key and value encoders ($\text{Encoder}^K$ and $\text{Encoder}^V$) are implemented as 3-layer MLPs with 1024 hidden units. We use OrientAnything [43] as the pre-trained direction key encoder ($\text{Encoder}^D$) and keep it frozen.

- **Model Size:** The total number of parameters in the model is around $488M$, of which $402M$ is trainable.

- **Rasterization:** We use the CUDA differentiable rasterizer implemented in the original 3DGS [16] to render the predicted 3D Gaussians. The rendering resolution matches the input resolution. We use near plane 0.1 and far plane 100.0 for rasterization.

**Training Configuration.** We employ a two-stage training strategy:

- **Optimizer:** We use AdamW [27] optimizer with learning rate $1e-4$, weight decay 0.05, and Cosine Annealing [26] learning rate schedule. The learning rate is warmed up for 2000 steps.
- **Training Losses:** The overall training objective (detailed in Section B) combines photometric and geometrical losses: $\mathcal{L}_{\text{total}} = \mathcal{L}_{\text{photo}} + \lambda_g \mathcal{L}_{\text{geo}} = \mathcal{L}_{\text{masked}} + \lambda_{\text{bg}} \mathcal{L}_{\text{bg}} + \lambda_g (\mathcal{L}_{\text{ray}} + \lambda_d \mathcal{L}_{\text{depth}})$.
- **Warm-up Training Stage:**
  - **Steps:** Trained for $250K$ steps with effective batch size 64. We train the model without the memory module during this stage.
  - **Loss Weights:** $\lambda_g = 0.3$, $\lambda_{bg} = 0.3$, $\lambda_d = 0.5$
  - **View-specific Embedding:** At this stage, we train the reference-view embeddings ($f_{ref}^{emb}$) and source-view embeddings ($f_{src}^{emb}$) while keeping the memory-view embeddings not involved.
  - **Initialization:** All weights are initialized randomly using the truncated normal distribution with mean 0 and standard deviation 0.02. Note that the pre-trained DINO encoder weights (i.e., $\text{Encoder}_1^I$) are frozen and not updated, while the learnable encoder ($\text{Encoder}_2^I$) is being updated.
  - **Input Sampling:** We sample $3-5$ views per sequence sample, with a sampling schedule as described in Sec. D.1.
- **Main Training Stage:**
  - **Steps:** Trained for $500K$ steps with effective batch size 16, incorporating the memory module.
  - **Loss Weights:** $\lambda_g = 0.3$, $\lambda_{bg} = 0.3$, $\lambda_d = 0.0$ ($\mathcal{L}_{\text{depth}}$ removed)
  - **View-specific Embedding:** To differentiate the two kinds of memory read-out tokens, we initialize two sets of memory-view embedding (i.e. $\{f_{mem1}^{emb}, f_{mem2}^{emb}\}$), one for orientation-aligned memory read-out and the other for orientation-complementary memory read-out.
  - **Initialization:** We initialize the transformer weights using the warm-up stage weights and we copy the source-view embeddings ($f_{src}^{emb}$) weights from the warm-up stage to initialize the memory-view embeddings ($\{f_{mem1}^{emb}, f_{mem2}^{emb}\}$). While the memory key encoder and value encoder weights are initialized randomly using the truncated normal distribution with mean 0 and standard deviation 0.02. Note that the pre-trained direction key encoder weights (i.e., $\text{Encoder}^D$) are frozen and not updated.
  - **Input Sampling:** We sample $6-12$ views per sequence sample, with a sampling schedule as described in Sec. D.1.

**Data Processing and Inference Details.**

- **Image Processing:** Input images are resized to $256 \times 256$ and normalized using ImageNet statistics [6]. We apply random augmentations including mirroring with a probability of 0.5.
- **Camera Normalization:** We preprocess all the ground truth poses in a sequence to be relative to the reference view, such that the reference view pose is the identity matrix.
- **Memory Sparsification:** The memory is pruned when reaching the maximum memory size, removing 20% of entries based on usage and coverage metrics. The temperature coefficient $\tau_t$ is dynamically adjusted based on orientation confidence: $\tau_t = 2.5 - \sigma_t$, where $\sigma_t \in [0, 1]$ is the inferred confidence from the orientation estimator [43].
- **Object Masking:** At inference time, we use the off-the-shelf video online segmentation model from Xmem [3] to estimate the object mask based on the initial object mask from the reference view.
- **Rendering:** At inference time, we filter out the predicted Gaussians that are either low in opacity (i.e., $o_k < 0.0001$) or the 0th-degree spherical harmonics coefficients are close to the background color (set to $[1, 1, 1]$ for rendering object against white background).
- **Training Environment:** We train our model on 8x NVIDIA A100 GPUs with 80GB memory. Our implementation uses Python 3.10, PyTorch 2.1.2, torchvision 0.16.2, and we leverage xFormers [19] 0.0.23 for efficient attention computation.
- **Evaluation Environment:** We run all inference on a single L40S GPU with 48GB memory.

# E Additional Discussions

## E.1 Additional Notes on Mesh Visual Comparison.

In Figure 4 of the main paper, we convert our final 3D Gaussian Splatting (3DGS) representation into meshes and conduct a comprehensive visual comparison with state-of-the-art methods across different paradigms. Methods

marked with an asterisk (*) indicate that additional pre- or post-processing steps were applied to generate the visual results. Below, we detail the mesh generation process for each method and provide additional analysis.

- For **Fmov** [31], **BundleSDF** [44], **Fast3R** [49], and **InstantMesh** [47], we either utilized their official implementations to generate meshes or obtained results directly from their published papers or official websites.

- **3DGS*:** Since the original 3DGS [16] implementation lacks support for pose-free object reconstruction and joint camera pose optimization, we employed the open-source gsplat [52] library with MCMC [17] random initialization for both 3DGS parameters and camera poses. We incorporated ground-truth object masks to optimize the 3DGS parameters.

- **COLMAP*:** We attempted reconstruction using the official COLMAP [30] implementation both with and without ground-truth object masks. However, neither approach converged successfully, resulting in failed reconstructions.

- **VGGT*:** Using the official VGGT [40] implementation, we preprocessed the input images by applying ground-truth object masks to remove background content. We utilized the Depthmap and Camera Branch outputs, which their paper indicates provide superior performance compared to the Pointmap branch. We additionally filtered out redundant background predictions to generate the final results.

- **Ours*:** To convert our OnlineSplatter framework's output into mesh format, we rendered 30 uniformly distributed views of the predicted 3D Gaussians, generating corresponding RGB-D images and masks. These were fused with their camera poses in a TSDF volume, and we extracted the mesh using Open3D [55]. The resulting mesh was further simplified using the open-source mesh simplification library [15]. Note that we only generate the mesh for a rough visual comparison, our method does not focus on mesh generation, we leave it as a future work.

From the visual comparison, our method demonstrates reconstruction quality comparable to approaches that require extensive optimization or additional depth supervision, while maintaining the advantages of a feed-forward, online framework. While optimization-based methods such as COLMAP* and 3DGS* excel at reconstructing static scenes, they face significant challenges when applied to freely moving objects, even with access to ground-truth object masks and the ability to perform global optimization across all frames. This observation highlights both the inherent difficulty of reconstructing freely moving objects from monocular videos and the promising capabilities of our approach. A promising future direction could be combining our online feed-forward framework with optimization-based refinement to achieve higher-quality mesh reconstructions.

## E.2 Broader Impacts

Our work on online 3D reconstruction has several societal implications. On the positive side, the technology could democratize 3D content creation by making real-time 3D scanning more accessible to the general public, while also improving efficiency in manufacturing through real-time quality control and inspection. The real-time nature of our system could benefit assistive technologies and education by enabling interactive 3D visualization and understanding of objects. However, there are potential concerns regarding privacy and intellectual property, as the ability to quickly reconstruct 3D models could be misused for unauthorized scanning or copying of physical objects. We recommend implementing appropriate usage guidelines and access controls when deploying the technology in sensitive contexts, while encouraging responsible development that considers both privacy and intellectual property rights.

