# OpenReview forum: "OnlineSplatter: Pose-Free Online 3D Reconstruction for Free-Moving Objects"
_NeurIPS.cc/2025/Conference — NeurIPS 2025 spotlight_

### Official Review · Reviewer_kmHg · 2025-06-25

**Clarity:** 2
**Significance:** 2
**Originality:** 2
**Rating:** 4
**Confidence:** 5

**Summary:**

The authors proposed a feed-forward reconstruction method in an online manner. This paper focuses on the reconstruction of free-moving objects. It introduces a dual-key memory model to fuse new features with the current state (object representation) via the Gaussian field. The authors claimed that such a spatial-guided memory readout keeps both the quality of object reconstruction and the computational cost. The authors also conducted a lot of experiments. However, as I asked in the following sections, I believe all of the experiments are about objects with rigid motions.

**Questions:**

Please refer to the weaknesses.

**Ethical Concerns:**

["NO or VERY MINOR ethics concerns only"]

**Final Justification:**

My concerns are addressed.

**Limitations:**

Please refer to the weaknesses.

**Paper Formatting Concerns:**

None.

**Quality:**

2

**Strengths And Weaknesses:**

**Strengths**
1. The authors introduced an online reconstruction method to reconstruct the free-moving objects.
2. The authors proposed a spatial-guided memory readout to achieve both reconstruction quality and computational efficiency.
3. The authors conducted extensive ablation studies.

**Weaknesses**
1. What is the difference between the Cut3r[1] and the proposed method? The Cut3r model also learn the state information. I will consider increasing my ratings if the authors can provide a reasonable answer.
2. This paper is limited to rigid motion. I think models like Cut3r and VGGT[2] can already tackle such tasks. So the problem settings and the motivation for this task are not obvious to me.
3. In L262, I highly suggest that the authors make their target clearer. I believe dynamic object reconstruction includes rigid motion and non-rigid motion. The authors should clarify which type their work belongs to.
4. In the experiments, I do believe that the quantitative experiments are not enough. I highly suggest that the authors conduct more comparisons with the current SOTA methods like Monst3r[3], Cut3r, and VGGT.

[1] Wang, Qianqian, Yifei Zhang, Aleksander Holynski, Alexei A. Efros, and Angjoo Kanazawa. "Continuous 3D Perception Model with Persistent State." arXiv preprint arXiv:2501.12387 (2025).

[2] Wang, Jianyuan, Minghao Chen, Nikita Karaev, Andrea Vedaldi, Christian Rupprecht, and David Novotny. "Vggt: Visual geometry grounded transformer." In Proceedings of the Computer Vision and Pattern Recognition Conference, pp. 5294-5306. 2025.

[3] Zhang, Junyi, Charles Herrmann, Junhwa Hur, Varun Jampani, Trevor Darrell, Forrester Cole, Deqing Sun, and Ming-Hsuan Yang. "Monst3r: A simple approach for estimating geometry in the presence of motion." arXiv preprint arXiv:2410.03825 (2024).

---

> ### Author Rebuttal · Authors · 2025-07-31
>
> We appreciate reviewer `kmHg`'s constructive feedback and recognizing our method achieves "**both reconstruction quality and computational efficiency**".
>
> > Q1: What is the difference between the Cut3r[1] and the proposed method? The Cut3r model also learn the state information. I will consider increasing my ratings if the authors can provide a reasonable answer.
>
> We thank the reviewer for this question regarding the recent work Cut3R[f]. While both our work and Cut3R[f] explore online feed-forward reconstruction, there are several fundamental differences in our approach and objectives.
>
> - Cut3R[f] tackles scene-level dynamic reconstruction, their output is a time-varying point cloud, whereas our method is object-level reconstruction, our output is a coherent object 3D representation. The difference in scope and final objective positions our paper quite differently.
> - The most critical distinction lies in how past information is aggregated. Cut3R[f]'s predictions are accumulative/additive, that is, it predicts new 3D points for each new frame without implicitly nor explicitly aggregating/aligning them with past predictions. That means, static regions in the scene will receive repetitive predicted points, and more importantly, dynamic regions (incl. moving objects) will receive multiple disconnected 3D instances of the same object in the output space. Specifically, if a rigid object is moving across the scene, the predictions of the object from Cut3R[f] will be scattered across the scene, creating multiple disconnected instances of the same object in the output space, and there is no way to aggregate them into a coherent object model. Thus, Cut3R[f] is unable to perform reconstruction for moving objects, which is the setting of our work.
> - Furthermore, due to the accumulative nature of Cut3R[f] in the output space, overlapping regions across views will receive repetitive predictions and they do not provide a principled way to align, nor remove those ever-growing unbounded number of points. Consequently, they also suffer growing memory usage, and significant drifting over time (as they acknowledged in their paper, and can be observed in their qualitative results).
> - In contrast, our method is implicitly aggregative for free-moving rigid objects. As we highlight in L132-L136 of our paper, our spatially-guided memory module is specifically designed to perform spatial coverage tracking, our dual-key memory readout strategy enables our OnlineSplatter transformer to learn to reason and align new observations with the past information at object level, such that we can discard out-dated past predictions. This enables our framework to produce a continuously-improving object representation with a fixed number of Gaussian primitives at every time step, effectively incorporating all past information while maintaining a constant memory footprint.
> - Cut3R[f] and our method both share the vision of incorporating a latent state to support online reconstruction, However, our method is tailored designed for object-level reconstruction. While Cut3R's prediction strategy serves well in understanding the dynamic scene, it does not serve well in our setting where we want to derive a continuously improving, well aligned 3D model for moving objects in online setting.
> - Having said that, if we try to adapt Cut3R[f] into our object-level settings and compare with them (although it is unfair to Cut3R and it's _not rigorous_ due to our fundamental differences in objectives), we extract the object region from per-frame predictions by Cut3R, and evaluate their object-level results in both per-frame mode (consider their per-frame output only) or accumulative mode (consider all past frame output combined). We provide the comparison on HO3D dataset and average across all frames and objects. Results show that our method outperforms Cut3R in both per-frame and accumulative mode.
>
> |        Method        | avg PSNR $\uparrow$ | avg SSIM $\uparrow$ | avg LPIPS $\downarrow$ | avg Chamfer Dist. $\downarrow$ |
> | :------------------: | :-----------------: | :-----------------: | :--------------------: | :----------------------------: |
> |  Cut3R (per-frame)   |       14.331        |        0.830        |         0.375          |             35.32              |
> | Cut3R (accumulative) |       14.760        |        0.882        |         0.419          |             59.26              |
> |         Ours         |       25.786        |        0.925        |         0.124          |              1.16              |
>
>
>
> > Q2: This paper is limited to rigid motion. I think models like Cut3r and VGGT[2] can already tackle such tasks. So the problem settings and the motivation for this task are not obvious to me.
>
> We appreciate the opportunity to clarify our problem setting and motivation.
> While Cut3R[f] & VGGT[30] are impressive recent works, they do not address the specific challenges of online reconstruction for freely-moving objects.
>
> VGGT[30] is designed for static scenes. When applied to video with a freely moving object, it implicitly treats the object as a transient outlier, causing it to be lost in the final output. We experimented with VGGT on HO3D test clips: feeding the original video (without object masks) to VGGT fails the reconstruction entirely. Moreover, even when we perform preprocessing and postprocessing using ground-truth object masks, VGGT still struggles to reconstruct the object, producing misaligned and redundant predictions (as we show in our Figure 4 of main paper). Furthermore, VGGT is stateless and does not handle online reconstruction or the continuous perception needed for real-time applications.
>
> Cut3R[f], as discussed in our response to Q1, handles moving objects by predicting separate point clouds for each frame. Without alignment or aggregation for moving objects, this results in multiple disconnected instances of the same object in the output space. This approach fails to leverage the continuous observation to build a single, coherent 3D object model, making it unusable in our object-level setting.
>
> Our work is motivated by the need for a continually improved 3D representation of a moving object, a realistic problem where an interacting agent (e.g. an autonomous robot) always observe an object partially first, and then interact with it, and gradually build up the complete understanding of the object. The continuous perception at object level is a problem that VGGT & Cut3R do not solve, while we achieve it with highly-efficient $O(1)$ memory-and-time update per time step.
>
> > Q3: In L262, I highly suggest that the authors make their target clearer. I believe dynamic object reconstruction includes rigid motion and non-rigid motion. The authors should clarify which type their work belongs to.
>
> We agree that the term "*dynamic object*" (in L262) can be ambiguous when viewed without context. To clarify, our work focuses on the online reconstruction of *freely-moving* ***rigid*** objects in dynamic environments, (see Line 84). Here, "*freely-moving*" refers to the object is undergoing arbitrary rotations and translations while a moving camera observes them, (see Line 22). This eliminates the common assumption often existing in object reconstruction works [cite a few] that the object must be static, centered in frame, and fully visible without occlusion.
> By alleviating these assumptions, our task becomes much more challenging and our method becomes much more applicable to real-world scenarios, for example, an autonomous robot observing a new object while handling it in a real household.
>
> To improve clarity, we will revise the term "dynamic object" in L262 to "freely-moving object" and emphasize the rigid object assumption in other relevant parts of the manuscript to ensure consistency and clarity.
>
> > Q4: In the experiments, I do believe that the quantitative experiments are not enough. I highly suggest that the authors conduct more comparisons with the current SOTA methods like Monst3r[3], Cut3r, and VGGT.
>
> We thank the reviewer for this suggestion. While MonST3R[e], Cut3R[f], and VGGT[30] are in related topics, they are designed for different tasks and settings than our work. A direct quantitative comparison would be challenging and potentially misleading, as the output and goals are not aligned. Specifically:
>
> - MonST3R[e] & VGGT[30] process a video as a whole in an offline setting, requiring all frames to be available upfront and fed into the model at once. It is not comparable to our online setting where video is fed as a stream, one frame at a time.
> - MonST3R[e] & Cut3R[f] focuses on scene-level dynamic environment modeling, understanding scene dynamics as a whole. They do not perform object-level reconstruction, and not easily adaptable to reconstruct free-moving objects.
> - In particular, Cut3R[f] outputs a per-step dynamic point cloud, but it does not aggregate object predictions into a coherent object model. Reference comparison is provided in Q1, but it is not rigorous due to our fundamental differences in objectives.
> - As a side note, both peer-reviewed papers of VGGT[30] & Cut3R[f] are only officially published on CVPR 2025 (released on June 3, 2025), after NeurIPS 2025 submission deadline. They are very recent, de facto concurrent works.
>
> However, we strongly agree that positioning our work relative to these methods is important. We will add a new discussion section in the paper that clearly articulates the differences in problem formulation, technical methodology, and objectives between our method and MonST3R[e], Cut3R[f], and VGGT[30]. We will also include qualitative comparisons where appropriate to visually highlight these distinctions and contextualize our contributions.
>
>
> [e] MonST3R: A Simple Approach for Estimating Geometry in the Presence of Motion (ICLR 2025)
>
> [f] Cut3R: Continuous 3D Perception Model with Persistent State (CVPR 2025)
>
> [30] VGGT: Visual Geometry Grounded Transformer (CVPR 2025)

---

> > ### Comment · Reviewer_kmHg · 2025-08-04
> >
> > Thanks to the authors for the rebuttal. Some of my concerns were addressed.
> >
> > However, it is essential to state the non-rigid motion feature of the reconstructed objects; otherwise, it will mislead the readers. Secondly, although Monst3r is not an online method, and its output is a point cloud, I do believe that more comparisons with Monst3r on depth (geometry) can be more convincing, since the authors claimed in their paper that the proposed method can capture the fine geometry details, and the depth map can be a good evaluation and cannot be affected by different reconstruction kernels. The NVS evaluations in Table 1 cannot concretely support the statements on geometry without depth evaluation, since the NVS metrics are actually not the geometry evaluation metrics. High NVS performance can also stay with bad geometry structures and details. Moreover, the authors only test the method on only **two** public datasets in the experiment section, which cannot convince me and also makes it hard to support a NeurIPS-accepted paper. Can I know how many objects in total have been tested in the paper? If there are no more public datasets like the ones in the paper, I highly suggest that the authors use similar scene datasets with masks. Thank you. I am looking forward to more discussions. Thank you.

---

> > > ### Author Response · Authors · 2025-08-05
> > >
> > > Thank you for your time and thoughtful review. We now address your additional comments:
> > >
> > > > However, it is essential to state the non-rigid motion feature of the reconstructed objects; otherwise, it will mislead the readers.
> > >
> > > Thank you for your constructive feedback. We will revise our manuscript accordingly to be more explicit about our setting and assumptions. We are willing to revise any claims that might have made you feel misled.
> > >
> > > > Secondly, although Monst3r is not an online method, and its output is a point cloud, I do believe that more comparisons with Monst3r on depth (geometry) can be more convincing, since the authors claimed in their paper that the proposed method can capture the fine geometry details, and the depth map can be a good evaluation and cannot be affected by different reconstruction kernels.
> > >
> > > We would like to clarify that the primary reason a direct comparison with MonST3R[e] is not feasible is that MonST3R[e] suffers from the same issue as Cut3R[f], where it produces multiple disconnected instances of the same moving object in the output space. Hence, there is no object-level representation to compare with. Moreover, MonST3R being an offline method, assuming all frames are available upfront. Although a forced comparison is possible, it would not do justice to either MonST3R[e] or our method.
> > >
> > > We also would like to clarify that we position our method as producing a **single, coherent, and continuously improving** 3D representation of a **freely moving** **rigid** object observed in a dynamic environment. Our method’s design is tailored to this setting; hence, we are able to produce more faithfully instance-specific geometry than methods like LRM[10], which leverage generative priors (more explanation can refer to our response to reviewer `bxd2`'s Q1), and methods like MonST3R[e], which do not implicitly or explicitly align the representation fo freely moving objects. We are willing to revise any claims that might have made you feel misled or over-claim.
> > >
> > > Here, we attempt to compare the object-region output of MonST3R[e] and our method using depth metrics. Specifically, for each frame, we consider the point depth predicted for the visible object region and normalize it by the maximum and minimum depth values of the object region ($d_{norm} = \frac{d - d_{min}}{d_{max} - d_{min}}$). We perform this normalization for our depth prediction and normalize the ground truth from camera view as well. We then compute the average depth $L1$ error over all frames. We provide the comparison on the HO3D dataset below:
> > >
> > > |   Method   | Avg. Normalized Depth Error $\downarrow$ |
> > > | :--------: | :--------------------------------------: |
> > > | MonST3R[e] |                  0.371                   |
> > > |    Ours    |                  0.366                   |
> > >
> > > Compared to MonST3R[e], our method achieves a comparable result. We believe this is expected when we only compare the per-frame object region **from current camera view**, as both MonST3R[e] and our method share the similar training signal of producing near-surface points or Gaussian primitives, the losses used in supervising the position of the 3D pointmap or the 3D Gaussian means are equivalent. Having said that, we would like to emphasize that this comparison does not reflect the core contribution of our method, as our main evaluation presented in the main paper **is evaluated from novel angles** to assess the object-level consistency and improvement over time in online setting.
> > >
> > >
> > > > The NVS evaluations in Table 1 cannot concretely support the statements on geometry without depth evaluation, since the NVS metrics are actually not the geometry evaluation metrics. High NVS performance can also stay with bad geometry structures and details.
> > >
> > > We appreciate the reviewer’s suggestion to add geometric accuracy metrics. In line with prior 3DGS-based reconstruction methods [2, 14, 39], we primarily reported reconstruction quality using NVS metrics. We will, however, add Chamfer Distance (CD) to better reflect geometric accuracy, as CD is more suitable and widely used in object-level settings. Below, we present CD results on the HO3D dataset, comparing against the FSO [36] and NPS [39] baselines, where we observe substantial improvements over both.
> > >
> > > |    Method     | Early-Stage CD $\downarrow$ | Mid-Stage CD $\downarrow$ | Late-Stage CD $\downarrow$ |
> > > | :-----------: | :-------------------------: | :-----------------------: | :------------------------: |
> > > | $FSO_{dist4}$ |            21.57            |           19.62           |           22.91            |
> > > | $NPS_{dist3}$ |            12.32            |           13.08           |           12.15            |
> > > |     Ours      |            1.31             |           1.14            |            1.03            |
> > >
> > > (continues below...)

---

> > > ### Author Response · Authors · 2025-08-05
> > >
> > > > Moreover, the authors only test the method on only two public datasets in the experiment section... Can I know how many objects in total have been tested in the paper?
> > >
> > > In evaluation, we used all **1032** unique real-world scanned objects from GSO[6] dataset, and all **11** unique real-world objects from HO3D[9] dataset. We evaluated on GSO due to its larger quantity of objects to better reflect the statistical significance of our results. We evaluated on HO3D due to its real-world captured video sequences containing challenging hand-object interactions and occlusions to better test the generalization of our method. Note that all objects used in our evaluation are **unseen** during our training.
> > >
> > > > If there are no more public datasets like the ones in the paper, I highly suggest that the authors use similar scene datasets with masks. Thank you. I am looking forward to more discussions. Thank you.
> > >
> > > We thank the reviewer for this suggestion. Scene datasets (even with object masks) are not easily applicable to our quantitative evaluation, as they do not have object-level representations or object-camera pose information for us to evaluate the object from novel angles, nor do they allow us to assess object-level consistency and improvement over time. However, we strongly agree with the reviewer that more experiments would be valuable in assessing the generalization of our method. We will adapt more samples from scene-level datasets and in-the-wild videos to our qualitative evaluation and add those results to our revised manuscript and appendix to further strengthen the paper.
> > >
> > > We sincerely appreciate your time and thoughtful feedback on our work. We have carefully considered and addressed your comments. If any points remain unclear, we would be more than happy to provide further explanation.

---

> > > > ### Comment · Reviewer_kmHg · 2025-08-06
> > > >
> > > > Most of my concerns are addressed; I will modify my ratings accordingly.

---

> > > > > ### Author Response · Authors · 2025-08-06
> > > > >
> > > > > Thank you for your positive re-evaluation. We are very pleased to hear that our clarifications have addressed your concerns. **We sincerely appreciate your time in reviewing our paper and rebuttal!**

---

### Official Review · Reviewer_bxd2 · 2025-06-28

**Clarity:** 3
**Significance:** 3
**Originality:** 4
**Rating:** 5
**Confidence:** 3

**Summary:**

This paper introduces a novel incremental and feed-forward pipeline for reconstructing freely moving objects from a sequence of images, with the final output represented as a 3D Gaussian Splatting (3DGS). The method is designed to address two central challenges: (1) performing reconstruction from unposed objects without depth priors, and (2) maintaining constant computational cost regardless of video length, enabling online applicability. While the first challenge has been explored in prior work, this paper contributes a practical and efficient solution to the second. Additionally, enabling feed-forward generation of 3DGS in an online setting is both technically demanding and impactful. Experiments on real-world datasets validate the effectiveness of the proposed method.

**Questions:**

Please refer to the Weaknesses.

In particular, including videos that showcase the 3D Gaussian Splatting reconstruction results along with their corresponding input video sequences would provide more convincing evidence of the method’s performance. I would be happy to raise my score if the above concerns are adequately addressed during the rebuttal.

**Ethical Concerns:**

["NO or VERY MINOR ethics concerns only"]

**Final Justification:**

1.The authors have thoroughly addressed my concerns in the rebuttal. I believe the paper meets the NeurIPS acceptance bar and have decided to increase my score to accept.

2.The work is complete and demonstrates novelty.

**Limitations:**

The authors have addressed the limitations of their work.

**Paper Formatting Concerns:**

No.

**Quality:**

3

**Strengths And Weaknesses:**

Strengths

1. This paper propose a novel feed-forward reconstruction pipeline characterized by a concise and effective design.

2. The introduced dual-key memory module demonstrates strong performance and effectively addresses practical challenge in incremental reconstruction.

3. The methodology is presented with clear and rigorous exposition, and the quantitative results convincingly validate the effectiveness of the approach.

Weeknesses

1. As stated in L30, the authors claim that prior methods struggle to capture instance-specific geometric details, limiting reconstruction accuracy. However, the proposed method does not appear to offer a clear advantage in capturing fine-grained geometry—apart from incorporating certain depth-based losses. This is particularly apparent when compared to 3DGS-based methods that benefit from explicit geometric initialization. Moreover, the authors do not report any quantitative metrics related to geometric accuracy (e.g., depth error, chamfer distance). It is therefore unclear whether the proposed method achieves better geometric fidelity or how it captures fine-grained geometry. If such improvement exists, further justification is needed. I would also encourage the authors to report geometric accuracy metrics to support their claim.

2. After masking out the background, the task essentially becomes pose-free object reconstruction. In this case, it would be more appropriate to compare against other pose-free baselines, such as COLMAP-Free 3D Gaussian Splatting. Although these methods are optimization-based, for 3D asset generation, reconstruction quality and geometric accuracy is often more critical than reconstruction efficiency. Such comparisons would help position the contribution more clearly.

3. The authors have not released the code, and the visualizations in the experiments are relatively limited. It is also unclear whether the test dataset includes objects that are entirely unseen in train dataset. The qualitative example of the "blue kettle" is not particularly convincing in terms of generalization. I would strongly encourage the authors to provide more diverse and challenging visual examples to support the claimed generalizability.

---

> ### Author Rebuttal · Authors · 2025-07-31
>
> We appreciate reviewer `bxd2`'s constructive feedback and recognizing our feed-forward generation of 3DGS in an online setting is "**both technically demanding and impactful**", the method is "**concise and effective**", our paper is "**clear and rigorous**". Below, we address reviewer's questions accordingly:
>
> > Q1: As stated in L30, the authors claim that prior methods struggle to capture instance-specific geometric details, limiting reconstruction accuracy. However, the proposed method does not appear to offer a clear advantage in capturing fine-grained geometry—apart from incorporating certain depth-based losses. This is particularly apparent when compared to 3DGS-based methods that benefit from explicit geometric initialization. Moreover, the authors do not report any quantitative metrics related to geometric accuracy (e.g., depth error, chamfer distance). It is therefore unclear whether the proposed method achieves better geometric fidelity or how it captures fine-grained geometry. If such improvement exists, further justification is needed. I would also encourage the authors to report geometric accuracy metrics to support their claim.
>
> We thank the reviewer for this question. We would like to clarify that, in Line 30, we specifically refer to the generative type of reconstruction methods, such as TRELLIS[d], LRM[10], and many more, that learn object-level priors via large-scale training, and **use hallucination as a feature**, with the main aim of *generating a plausible 3D asset* even when a large portion of the 3D geometry is not observed. These methods are great for single-image-to-3D asset generation, but if we adapt them to online perception settings, not only does their hallucination feature not help in constructing faithful “instance-specific geometry,” but their predictions also do not benefit from continuous observation (i.e., stream-based video input), which should improve the object reconstruction over time, as we demonstrated in our work.
>
> In order to reconstruct object faithfully to the instance on the fly while observing it, we leverage recent advances in training feed-forward models using 3DGS parameterization (e.g. pixelSplat[2], NoPoSplat[39]), where our primary training signal teaches the model to produce pixel-aligned Gaussian primitives on the observable object surface. Different from the generative line of works that we mention above, our training objective is implicitly different, where our depth-based and ray-alignment losses supervises the model to predict local geometric details, rather than object-level distributions. Therefore, to reconstruct the full object, we designed the spatial-guided object memory to perform spatial coverage tracking, and the dual-key memory readout strategy to fuse new observations into a single, canonical 3D object representation. All of these designs ensures that our model reconstruct faithfully to the instance, and the continuous observation enables our model to improve the reconstruction over time.
>
> We appreciate the reviewer’s suggestion to add geometric accuracy metrics. In line with prior 3DGS-based methods [2, 14, 39], we primarily reported reconstruction quality using PSNR, SSIM, and LPIPS. We will, however, add Chamfer Distance (CD) to better reflect geometric accuracy. Below, we present CD results on the HO3D dataset, comparing against the FSO [36] and NPS [39] baselines, where we observe substantial improvements over both.
>
> |    Method     | Early-Stage CD $\downarrow$ | Mid-Stage CD $\downarrow$ | Late-Stage CD $\downarrow$ |
> | :-----------: | :-------------------------: | :-----------------------: | :------------------------: |
> | $FSO_{dist4}$ |            21.57            |           19.62           |           22.91            |
> | $NPS_{dist3}$ |            12.32            |           13.08           |           12.15            |
> |     Ours      |            1.31             |           1.14            |            1.03            |
>
>
> [d] TRELLIS: Structured 3D Latents for Scalable and Versatile 3D Generation (CVPR 2025)
>
> > Q2: After masking out the background, the task essentially becomes pose-free object reconstruction. In this case, it would be more appropriate to compare against other pose-free baselines, such as COLMAP-Free 3D Gaussian Splatting. Although these methods are optimization-based, for 3D asset generation, reconstruction quality and geometric accuracy is often more critical than reconstruction efficiency. Such comparisons would help position the contribution more clearly.
>
> We appreciate the reviewer's perspective, however, we do not position our paper for 3D asset generation. We agree that for 3D asset generation, efficiency doesn’t matter that much given that existing methods can already produce 3D assets in less than a few seconds. Instead, the value added by our novel online formulation comes from downstream applications that require real-time 3D perception and continuous perception, in such applications, it values efficiency and bounded memory footprint. Therefore, we tackle the challenging balance between compute-memory efficiency and good reconstruction quality.
>
> Additionally, in the Figure 4 of our main paper, we do show that using COLMAP-Free 3DGS with Markov Chain Monte Carlo (MCMC) does not work out of the box for (free-moving rigid) object reconstruction, where we also added joint pose optimization and employ ground-truth object masks on top of COLMAP-Free 3DGS to perform global optimization using all frames. despite extended optimization (over 6 minutes on a 4090 GPU), the end result could still be very unsatisfactory (with messy floaters & inaccurate geometry).
>
> While dedicated offline methods like GaussianObject[c] can achieve higher quality reconstruction, they do so through a complex multi-stage optimization pipeline, including computing visual hull -> optimize a coarse 3DGS model -> perform leave-one-out training -> fine-tune a LoRA for repair -> perform Gaussian repair, as detailed in [c], placing them in a different category from our online approach that can do on-the-fly reconstruction.
>
> [c]: GaussianObject: High-Quality 3D Object Reconstruction from Four Views with Gaussian Splatting (TOG 2024)
>
>
> > Q3: The authors have not released the code, and the visualizations in the experiments are relatively limited. It is also unclear whether the test dataset includes objects that are entirely unseen in train dataset. The qualitative example of the "blue kettle" is not particularly convincing in terms of generalization. I would strongly encourage the authors to provide more diverse and challenging visual examples to support the claimed generalizability.
>
> We thank the reviewer for these important points. We will release our code and pre-trained models upon acceptance to facilitate reproducibility. We can also confirm that our training and evaluation datasets (Objaverse and GSO/HO3D, respectively) are entirely separate, all test objects were **unseen during training**.
>
> We also agree that more diverse visualizations would strengthen the paper. Unfortunately, NeurIPS policy prevents us from providing additional visual materials or external links during the rebuttal period, we will add more challenging qualitative examples to the technical appendix and a project website to better demonstrate the generalization capabilities of our method.
>
> > Q4: In particular, including videos that showcase the 3D Gaussian Splatting reconstruction results along with their corresponding input video sequences would provide more convincing evidence of the method’s performance. I would be happy to raise my score if the above concerns are adequately addressed during the rebuttal.
>
> We thank the reviewer for this excellent suggestion. Unfortunately, NeurIPS policy prevents us from providing additional visual materials or external links during the rebuttal period, we fully commit to addressing this concern upon acceptance. Specifically, we will showcase more progressive qualitative results (like our Figure 1) in our technical appendix and provide more video-based demo on our website.

---

> > ### Comment · Reviewer_bxd2 · 2025-08-01
> > **Response**
> >
> > Thank you very much for your response — it addressed my concerns thoroughly. I believe the paper meets the NeurIPS acceptance threshold, and I will update my score accordingly.
> >
> > I also noticed Reviewer kmHg's comments. In my view, the distinction between photorealistic novel view synthesis (e.g., NeRF/3DGS) and general-purpose reconstruction methods (e.g., COLMAP, VGGT, MVS) may have been overlooked. While outputs from systems like VGGT could potentially serve as inputs to pipelines such as 3DGS, the two tasks have fundamentally different goals. I do not think a direct comparison is entirely appropriate, and I plan to raise this point in the discussion with the AC and other reviewers. Personally, I would be very interested in trying the method once the code is released.
> >
> > Lastly, I have a minor technical question: since Chamfer Distance is defined between two point clouds, how do you obtain both the predicted and ground-truth point clouds for evaluation?

---

> > > ### Author Response · Authors · 2025-08-01
> > >
> > > We thank the reviewer for the positive comments, we also appreciate the comments pointing out that a direct comparison with certain works such as VGGT is not entirely appropriate.
> > >
> > > To answer your technical question: Ground-truth point clouds are often sampled from ground-truth object model, and pre-sampled point cloud is also available for evaluation. For predicted point cloud in our case, for simplicity, we acquire the XYZ coordinates from our predicted 3D Gaussian primitives (i.e. the Gaussian means) as the predicted point cloud.

---

> ### Comment · Reviewer_bxd2 · 2025-08-01
>
> Thank you for your response. I have updated(raised) my score accordingly.

---

### Official Review · Reviewer_ZqiV · 2025-07-02

**Clarity:** 3
**Significance:** 2
**Originality:** 3
**Rating:** 5
**Confidence:** 4

**Summary:**

The paper proposes a feed-forward method for object tracking and reconstruction in an online manner. The method operates only on RB frames and claims to be real-time. Moreover, the proposed method doesn't require any camera poses and references everything w.r.t. to the canonical representation in the first frame. Experimental validation is sufficient and shows new state-of-the-art performance on several datasets.

**Questions:**

1. Would a comparison to [8] be possible?
2. Another ablation on the used VOS module would be interesting.
3. Is it possible to extend the training procedure to avoid requiring ground truth camera poses?
4. The method doesn't use any global optimisation, but it can benefit from it, e.g. 3DGS optimization on all frames. How much would this improve the results?

**Ethical Concerns:**

["NO or VERY MINOR ethics concerns only"]

**Final Justification:**

Most of my concerns are well-addressed, and I hope that those additional experiments will be added to the final paper (including ablating different VOS modules, comparison to [8], etc). I'm willing to raise my score and view paper as worth publishing at NeurIPS.

**Limitations:**

Yes, the limitations are discussed in the supplementary materials.

**Paper Formatting Concerns:**

No.

**Quality:**

3

**Strengths And Weaknesses:**

Strengths:
* The proposed method is novel, and the idea of using a feed-forward model for splatting a dynamic object reconstruction is interesting.
* The whole pipeline seems to be well-engineered, e.g. runs in real-time, operates in an online manner, and requires only monocular RGB frames.
* The method achieves state-of-the-art results on GSO and HO3D datasets, both in early, mid and late stage of object tracking. The improvement over other methods is significant.
* The introduced dual-key memory module is an important contribution, together with the memory sparsification mechanism that drops tokens that are least useful.

Weaknesses:
* The method assumes that object masks as already known. This requires some user input and may limit some applications since it requires some user clicks and the video object segmentation model XMem.
* Training still requires ground-truth poses, and overall the training is very supervised. This could be a problem for good generalization capabilities of this method.
* The paper should also compare to other methods, such as [8].
* Eq. (9) is confusing. Why is there a dot sign followed by a minus sign? Are brackets missing there?
* There are many typos, missing letters, etc. The paper would benefit from proof-reading.

---

> ### Author Rebuttal · Authors · 2025-07-31
>
> We appreciate reviewer `ZqiV`'s constructive feedback and recognizing our "**method is novel**", the idea is "**interesting**", pipeline is "**well-engineered**". Below, we address reviewer's questions accordingly:
>
> > Q1: The method assumes that object masks as already known. This requires some user input and may limit some applications since it requires some user clicks and the video object segmentation model XMem.
>
> Yes, in order for our current method to run, we currently require an initial specification of the object of interest. This is because there may be multiple objects in the video, and so some human input is needed to specify the target object at test time. The current setup is consistent with related works[33] in tracking and reconstruction in online settings.
>
> Crucially, after this initial step, our online pipeline leverages the highly efficient XMem Video Object Segmentation (VOS) module. XMem is well-suited for online applications due to its strong performance, speed, and bounded memory footprint. Therefore, the computational overhead is minimal and primarily confined to the first frame, ensuring our framework remains practical for a wide range of real-time applications.
>
> Furthermore, the input mechanism is flexible (e.g., click, box, or text prompt) and not a hard constraint of the architecture. For applications requiring full automation, our framework can also be readily extended to process all salient foreground objects in parallel by integrating an off-the-shelf object detector.
>
>
> > Q2: Training still requires ground-truth poses, and overall the training is very supervised. This could be a problem for good generalization capabilities of this method.
>
> We agree with the reviewer that the current training setup requires ground-truth poses, however, we would like to clarify that, firstly, these ground-truth poses are obtained at nearly no cost. As long as we have an object model, either a virtual object model, a real-world scanned model, or a generated one, we can render it at unlimited views and obtain RGB frames with ground-truth poses. Thus, we do not face the issue of data scarcity in terms of camera poses required by our method.
>
> Furthermore, to ensure robust generalization, we took several deliberate steps in our training. We generated our training data using a custom Blender script, which introduces diverse and randomized camera trajectories, lighting, and focal lengths (as detailed in our Supp. D.2). This strategy minimizes distributional bias and exposes the model to a comprehensive range of relative object-camera poses, such that our learned model is not biased towards a specific range of relative position. Our evaluations validates our generalization capabilities on datasets with entirely unseen object categories, including challenging real-world data from HO3D, demonstrate the success of this approach.
>
> Having said that, the idea of removing the reliance on ground-truth poses in training is an interesting and challenging problem to solve. Currently, the ground-truth poses are essential due to our reliance on using 3DGS differentiable rendering to provide gradient from RGB frames to model parameters. We could consider treating the camera pose as a latent variable and optimize reconstruction losses (photometric/silhouette) after marginalizing over poses proposed by a small amortized pose predictor. We could also leverage feedback from external discriminator (a judging model to compare predicted and ground-truth frames) as a training signal to guide the pose prediction, but we leave this interesting direction as future work.
>
>
> > Q3: The paper should also compare to other methods, such as [8].
>
> We thank the reviewer for this suggestion. [8] is an inspiring and seminal work that tackles free-moving object reconstruction via multi-stage offline bundle optimization. The primary reason for not including a direct quantitative comparison with [8] is the fundamental difference in our objectives and assumptions. Our work focuses on an online, feed-forward framework designed for continuous perception, whereas [8] is an offline, bundle optimization-based method, assuming all frames are available upfront. Furthermore, lacking of open-source code from [8] also made the comparison extremely challenging.
>
> While it is not a fair comparison for either work, below, we try to compare with [8] by taking numerical results directly from their paper, and we report our reconstruction result at the final time step. We evaluate on the same objects as described in [8]'s paper and we calculate the Hausdorff distance following [8]. The results shown that our method achieves better reconstruction quality than [8] in terms of Hausdorff distance, this is consistent with our qualitative results in Figure 4 when compare with qualitative results of [8].
>
> | Method | avg. Hausdorff dist. |
> | :----: | :------------------: |
> |  [8]   |         4.65         |
> |  Ours  |        2.196         |
>
> We are unable to compare the PSNR/SSIM/LPIPS metrics due to the lack of their evaluation details, we are unable to infer whether their reported results are computed via rendering or texture map. However, we will add a visual comparison with the results presented on [8]'s official website to our Figure 4. (We are unable to provide the link nor the visualization here due to NeurIPS policy)
>
> > Q4: Eq. (9) is confusing. Why is there a dot sign followed by a minus sign? Are brackets missing there?
>
> Yes, thanks for the suggestion, adding a pair of bracket would imporove clarity, we will update Eq.(9) as: $s_i^{(\text{comp})} = (\mathbf{q}_t^{(L)\top}\mathbf{k}_i^{(L)}) \cdot (-\mathbf{q}_t^{(D)\top})\mathbf{k}_i^{(D)} \cdot \tfrac{1}{\tau_t}$
>
> > Q5: There are many typos, missing letters, etc. The paper would benefit from proof-reading.
>
> Thank you, we will thoroughly proof‑read and fix any typos.
>
> > Q6: Would a comparison to [8] be possible?
>
> Thank you for the question. Please see our detailed response to Q3.
>
> > Q7: Another ablation on the used VOS module would be interesting.
>
> This is an insightful suggestion. We selected XMem[3] as our default VOS module due to its well-established performance, efficiency, and bounded memory usage, which aligns perfectly with our online framework's requirements.
>
> For our experiments, we pre-processed all object masks to ensure a fair and rigorous comparison between our method and the baselines, eliminating segmentation as a variable. However, to address the reviewer's question, we conducted a brief analysis using other state-of-the-art VOS models, such as Cutie[a] and SAM-2[b]. We observed that these models produce highly comparable segmentation results on our test set, with performance variations leading to negligible differences (<0.5%) in our final reconstruction metrics. This suggests our pipeline is not overly sensitive to the choice of a specific high-performing VOS module, thus we selected XMem[3] as our default VOS module due to its smaller model size compared to the other state-of-the-art VOS models.
>
> [a]: Cutie: Putting the Object Back into Video Object Segmentation (CVPR 2024)
> [b]: SAM 2: Segment Anything in Images and Videos (2024)
>
>
> > Q8: Is it possible to extend the training procedure to avoid requiring ground truth camera poses?
>
> Thank you for the question. Please see our detailed response to Q2.
>
> > Q9: The method doesn't use any global optimisation, but it can benefit from it, e.g. 3DGS optimization on all frames. How much would this improve the results?
>
> This is an excellent question regarding the potential benefits of post-process optimization. We agree that incorporating a global optimization on all frames would improve the final reconstruction quality. However, we conducted such experiments where we leverage 3DGS + joint pose optimization and implemented additional losses to encourage pixel alignment (similar to our training loses), we are only able to get an average $+1.26$ PSNR performance gain by optimizing for more than 100,000 iterations or over 2 mins.
>
> We attribute the underwhelming improvement over prolonged time to a few reasons: 1. 3DGS optimization is sensitive to hyperparameter choice (such as different learning rate and scheduler are required for different parameter groups, especially for joint pose optimization, requires careful tuning), while our feed-forward model already produces good results from large-scale training, added per-instance optimization may struggle to improve further. 2. 3DGS optimization is also hyper-sensitive to imperfect object masks and it is prone to produce floaters around object edges, we have shown this evidence in our Figure 4 of main paper, the floaters around the edges significantly harms the performance. While our feed-forward method can be trained to handle the object edges and imperfections in object masks better by batched training.
>
> However, we believe there is still room for exploration in this hybrid direction, by carefully integrating optimization into our online framework while maintaining the efficiency and bounded memory footprint, we leave this as future work.

---

> > ### Comment · Reviewer_ZqiV · 2025-08-04
> >
> > Thank you for your answers. Most of my concerns are well-addressed, and I hope that those additional experiments will be added to the final paper (including ablating different VOS modules, comparison to [8], etc). I'm willing to raise my score and view paper as worth publishing at NeurIPS.
> >
> > After reading other reviewers, I agree with reviewer bxd2 that reviewer kmHg's concerns about CUT3R and VGGT are overstated.

---

> > > ### Author Response · Authors · 2025-08-04
> > >
> > > Thank you for your positive comments and re-evaluation. We are very encouraged. We are also very pleased to hear that our clarifications and additional results have addressed your concerns.
> > >
> > > We appreciate your comments that “reviewer kmHg's concerns about CUT3R and VGGT are overstated.”
> > >
> > > Your feedback was invaluable and has significantly helped us improve the quality and clarity of our manuscript. We sincerely appreciate your time in reviewing our paper and rebuttal.

---

### Decision · Program_Chairs · 2025-09-17

**Decision:**

Accept (spotlight)

**Comment:**

All reviewers recommend accepting this paper, congratulations!

The paper proposes a novel feed-forward pipeline for dynamic object reconstruction with a dual-key memory module and sparsification, enabling real-time online performance from monocular RGB input. The approach is well-designed, achieves strong improvements on GSO and HO3D, and is methodologically distinct from prior work such as VGGT/CUT3R.

Reviewers highlighted the paper’s novelty, clarity, and empirical strength, but also noted limitations: reliance on known masks and supervised training, lack of geometric fidelity metrics, and missing comparisons to pose-free 3DGS baselines. Additional qualitative results, improved visualizations, and code release would further strengthen the work. Although I can see the motivation for questioning around the work on VGGT/ CUT3R, the problem setup and method itself is different enough. In addition, authors provided good rebuttal contents that is fairly convincing. Please include some of these discussion in the final version.

Overall, the rebuttal addressed key concerns and the strengths outweigh the weaknesses. I recommend acceptance, and encourage the authors to incorporate reviewer feedback—particularly on experimental completeness, geometric evaluation, and clarity. Given its novelty and potential applications, this paper is also a good candidate for a highlight presentation.